# The Influence of Simulated Surface Dust Lofting and Atmospheric Loading on Radiative Forcing

Stephen M. Saleeby[1], Susan C. van den Heever[1], Jennie Bukowski[1], Annette L. Walker[2], Jeremy E. Solbrig[3], Samuel A. Atwood[1], Qijing Bian[1], Sonia M. Kreidenweis[1], Yi Wang[4], Jun Wang[4], Steven D. Miller[3]

[1]Department of Atmospheric Science, Colorado State University, Fort Collins, Colorado
[2]Naval Research Laboratory, Monterey, California
[3]Cooporative Institute for Research in the Atmosphere, Fort Collins, Colorado
[4]Department of Chemical and Biochemical Engineering, Interdisciplinary Graduate Program in Informatics, and Center for Global and Regional Environmental Research, University of Iowa, Iowa City, Iowa

*Correspondence to:* Stephen M. Saleeby (Stephen.Saleeby@colostate.edu)

**Abstract.** This high-resolution numerical modeling study investigates the potential range of impact of surface-lofted dust aerosols on the mean radiative fluxes and temperature changes associated with a dust lofting episode over the Arabian Peninsula (2-5 August 2016). Assessing the potential for lofted dust to impact the radiation budget and temperature response in regions of the world that are prone to intense dust storms is important due to the impact of such temperature perturbations on thermally driven mesoscale circulations such as sea-breezes and convective outflows. As such, sensitivity simulations using various specifications of dust erodible fraction were performed using two high-resolution mesoscale models that use similar dust lofting physics based on threshold friction wind velocity and soil characteristics. The dust erodible fraction, which represents the fraction (0.0 to 1.0) of surface soil that could be mechanically lifted by the wind and controls the location and magnitude of surface dust flux, was varied for three experiments with each model. The "Idealized" experiments, which used an erodible fraction of 1.0 over all land grid cells, represent the upper limit on dust lofting within each modeling framework, the "Ginoux" experiments used a 1-degree resolution, spatially-varying erodible fraction dataset based on topographic depressions, and the "Walker" experiments used satellite-identified, 1-km resolution data with known lofting locations given an erodible fraction of 1.0. These simulations were compared to a "No-Dust" experiment in which no dust aerosols were permitted. The use of erodible fraction databases in the Ginoux and Walker simulations produced similar dust loading which was more realistic than that produced in the Idealized lofting simulations. Idealized lofting in this case study generated unrealistically large amounts of dust compared to observations of aerosol optical depth (AOD), due to the lack of locational constraints. Generally, the simulations with enhanced dust mass via surface lofting experienced reductions in daytime insolation due to aerosol scattering effects, as well as reductions in nighttime radiative cooling due to aerosol absorption effects. These radiative responses were magnified with increasing amounts of dust loading. In the Idealized simulation with "extreme" (AOD > 5) dust amounts, these radiative responses suppressed the diurnal temperature range. In

the Ginoux and Walker simulations with "moderate" (AOD ~ 1-3) amounts of lofted dust, the presence of dust still strongly impacted the radiative fluxes but only marginally modified the low-level temperature. The dust-induced near-surface temperature change was limited due to competing thermal responses to changes in the net radiative fluxes and the dust layer radiative heating rates. Compared to the Ginoux simulation, the use of increased resolution in dust erodible fraction inventories in the Walker simulations led to enhanced fine-scale horizontal variability in lofted dust and a modest increase in the mean dust concentration profile and radiative/thermal responses. This study discusses the utility of using high-resolution dust source databases for simulating lofted dust, the need for greater spatial coverage of in situ aerosol observations in dust prone regions, the impacts of dust on the local radiation budget and surface thermal conditions, and the potential dust radiative impacts on thermally driven mesoscale features.

## 1 Introduction

Dust aerosols are a substantial contributor to the global aerosol population, particularly in the dust belt region (Prospero et al. 2002; Tanaka and Chiba, 2006). They are known to strongly influence the radiation budget due to their significant scattering and absorption properties (Carlson and Benjamin, 1980; Haywood et al., 2003; Kinne et al., 2003; Sokolik and Toon, 1996; Dubovik et al., 2006). Dust aerosol layers can contribute to low-level atmospheric cooling due to the attenuation of shortwave radiation (solar dimming) through both scattering and absorption at solar wavelengths (Carlson and Caverly, 1977; Tegen and Lacis, 1996; Slingo et al., 2006; Lau and Kim, 2007; Kosmopoulos et al., 2017). Solar dimming can also lead to reduced surface heating, and thus, reduced latent and sensible heat fluxes (Wang et al., 2004; Prakash et al., 2015). In contrast, dust absorption of both longwave and shortwave radiation can contribute to localized heating by directly warming the dust-laden atmospheric layer and increasing downward thermal emission and by reducing the amount of surface thermal emission escaping to space (Tegen and Lacis, 1996; Slingo et al., 2006; Lau and Kim, 2007). A cooling tendency within a dust layer may also exist due to longwave emission, with a warming tendency adjacent to the dust layer (Slingo et al., 2006; Wang et al., 2013). The vertical distribution of dust also exerts a strong influence over surface and low-level radiative forcing and temperatures by modifying the vertical locations of solar scattering and radiative heating/cooling (Tegen and Lacis, 1996; Hsu et al., 2000; Slingo et al., 2006, Sokolik and Toon, 1996; Lau and Kim, 2007). A combination of the vertical distribution of dust, the overall aerosol loading, and the complex balance among shortwave scattering and absorption and longwave absorption and emission determines the net impact of dust on the low-level tropospheric temperature profile. As such, much uncertainty and variability remain among studies focused on the overall thermodynamic impact of dust storms with marked variability found on a case by case basis and with respect to varying observational and modeling platforms (Tegen and Lacis, 1996; Slingo et al., 2006; Prakash et al., 2015).

Through aerosol absorption and scattering processes associated with aerosol optical properties, lofted dust that is concentrated near the surface could potentially impact the lower atmospheric radiation budget sufficiently enough to alter the daytime and nighttime surface heating and low-level temperature profiles. Modification of the thermal profiles by dust

loading has implications on the development of mesoscale weather features such as convection and sea-breezes whose circulations are driven or initiated by horizontal heterogeneities in local-scale thermal contrasts (e.g., Eager et al., 2008; Verma et al., 2006; Crosman and Horel, 2010; Ge et al., 2014). As such, it is necessary to improve our understanding of how dust lofted from Earth's surface can impact radiative quantities. Simulating dust lofting and its direct radiative effects via appropriate aerosol and radiation parameterizations within a numerical weather prediction model provides an effective way of elucidating the impacts of dust aerosols on the components of radiative and surface heating, and ultimately, their impact on low-level atmospheric temperature.

Numerical modeling of dust lofting requires a mechanical method of lofting dust from the surface dictated by the surface wind threshold friction velocity (Westphal et al., 1988; Marticorena and Bergametti, 1995). Dust lofting potential also varies with the soil type (Fecan et al., 1998) and vegetation (Pierre et al., 2012) and can be represented using geographical location datasets of dust erodible fraction that may vary dramatically in spatial coverage and resolution (Ginoux et al., 2001; Walker et al., 2009). The erodible fraction represents the percentage of surface soil that could potentially be mechanically lifted by the wind. Lofting by strong surface winds is favored in dry areas with bare, loose soil, and little to no vegetation. Inventories of dust erodible fraction (e.g. Ginoux et al., 2001; Walker et al., 2009) can be interfaced with dust lofting parameterizations to constrain the dust lofting potential over a given region.

Dust lofting occurs frequently over Saudi Arabia and along the Persian Gulf coastlines of the Arabian Peninsula (Tanaka and Chiba, 2006; Eager et al., 2008) and is maximized during the summer months (Prospero et al., 2002; Goudie and Middleton, 2006). Even though the Arabian Peninsula contributes substantially to the total lofted dust load in the northern Hemisphere (Tanaka and Chiba, 2006), few studies have focused on this region compared to the Sahara and east Asia (Prakash et al., 2015). Sea-breezes occur frequently along the Arabian Peninsula coastal zones, resulting from the intense heating of the land and strong land/sea temperature gradient (Verma et al., 2006; Eager et al., 2008), and dust lofting in this region is also maximized during the daytime due to enhanced local scale winds and turbulence associated with daytime heating (Middleton et al., 1986), as well as dust lofted by convective outflows (Miller et al., 2008). As such, this is a prime location for simulating and examining dust lofting in terms of its potential radiative impacts on the regional/local scale temperatures and associated forcing of mesoscale circulations.

This paper seeks to address research goals outlined by the Holistic Analysis of Aerosols in Littoral Environments (HAALE) team, a Multidisciplinary Research Program of the University Research Initiative (MURI) operating under auspices of Office of Naval Research (ONR). An overarching objective of the research is to identify the fundamental environmental factors that govern the spatial distribution and optical properties of littoral zone aerosols (including dust) at the sub-km scale. Within this scope, we hope to advance our understanding of aerosol direct and indirect impacts on the littoral zone meteorology, optical depth, and visibility and their associated feedbacks. As such, this paper seeks to first examine the

predictability of dust generation and transport in models, and then, determine the influence of these predictions on the radiation budget in terms of feedbacks to the atmospheric thermodynamic structure. The HAALE team has chosen case studies that involve regions of intense aerosol production and transport that could interact with these littoral zone processes.

Numerical simulations were performed for a dust lofting event that occurred over the Arabian Peninsula from 2-5 August 2016. In this event, dust was lofted over this very arid region from multiple locations and multiple directions via strong surface winds. Simulations were performed using both the Regional Atmospheric Modeling System (RAMS) (Cotton et al., 2003; Saleeby and van den Heever, 2013) and the Weather Research and Forecasting Chemistry Model (WRF-Chem) (Skamarock et al., 2008; Grell et al., 2005; Fast et al., 2006). These two modeling frameworks were used to determine if

there is a consistent and robust parameterization representation of dust emission and its impact on radiation parameters and temperature profiles. Within each model framework (RAMS and WRF-Chem), the analysis focuses on the following aspects: (1) the variability in lofted dust amounts from three different methods of specifying the dust surface erodible fraction, (2) the direct radiative impacts of the predicted dust lofting on the atmospheric and surface heating, and (3) the ultimate influence of these variations on the surface diurnal temperature cycle and atmospheric temperature profile.

The paper is outlined as follows: Sections 2.1 and 2.2 detail the RAMS and WRF-Chem model simulation designs and relevant parameterizations, respectively. Section 2.3 provides information on the three dust erodible fraction specifications being tested in this study. Section 2.4 describes the synoptic background setup for the 2-5 August 2016 Arabian Peninsula dust event. Section 3 provides simulated dust lofting and aerosol optical depth comparisons, and Section 4 details the dust

radiative and temperature impacts. Section 5 concludes the paper with a summary of the main findings.

## 2 Model and Case Study Descriptions
### 2.1 RAMS Model Specifications
The RAMS model (Pielke et al., 1992; Cotton et al., 2003) version 6.2.06 was run over the Arabian Peninsula and

surrounding region (Figure 1). This open-source version of RAMS is currently maintained by the research group of Prof. Susan C. van den Heever of the Department of Atmospheric Science at Colorado State University and can be found at the following URL: https://vandenheever.atmos.colostate.edu/vdhpage/rams.php. Initial RAMS simulations were run from 0000 UTC $2^{nd}$ of August 2016 to 0000 UTC $5^{th}$ of August 2016 on Grid-1 (15 km grid spacing) (Fig. 1a). The 1-degree gridded Global Data Assimilation System Final Analysis (GDAS-FNL) data at 6-hour intervals were used to initialize and provide

lateral boundary nudging for RAMS Grid-1. This parent grid (Grid-1) simulation was then used to generate the initial and boundary conditions for the Grid-2 simulations (2 km grid spacing) run from 0000 UTC Aug 3 for 48 hours with model analyses available at 30-minute intervals (Fig. 1b). Both simulations were run with 50 terrain-following sigma-z vertical levels on a stretched grid with minimum vertical grid spacing of 75 m near the surface. A summary of the RAMS model configuration and a general description of the physics packages used for simulations in this study are given in Table 1.

The RAMS double-moment microphysics parameterization module predicts the number concentration and mass mixing ratio of three liquid and five ice hydrometeor species (Walko et al., 1995; Meyers et al., 1997). Aerosol activation and cloud droplet nucleation are parameterized according to Saleeby and Cotton (2004) and Saleeby and van den Heever (2013). Aerosol particles may be scavenged through nucleation and wet and dry deposition (Saleeby and van den Heever, 2013). Dust aerosols are mechanically lofted from the surface using the methods being tested herein (described below), and sea salt aerosols are generated over ocean surfaces under windy conditions as described in Saleeby and van den Heever (2013). Finally, an initial background pollution aerosol population was applied with a clean-continental profile containing 600 cm$^{-3}$ at the surface and reduced exponentially aloft, similar to the clean-continental aerosol number concentration profile in Saleeby et al. (2016). All aerosol species can scatter and absorb shortwave and longwave radiation, thus, providing feedbacks to the dynamics and thermodynamics (Harrington, 1997; Stokowski, 2005). The refractive indices used across wavelengths for each aerosol species are shown in Stokowski (2005) and are guided by field data from the Saharan Dust Experiment (SHADE) (Haywood et al., 2003). For the radiative parameterization of dust species in RAMS an index of refraction of 1.53+0.0015i was used up to ~2000nm wavelength for building the optical lookup tables. The refractive index varied at longer wavelengths (Stokowski, 2005). Use of these values produced dust layer heating comparable to observations from SHADE (Stokowski, 2005).

Mechanical dust lofting in RAMS is a function of the threshold friction wind velocity (Marticorena and Bergametti, 1995), clay fraction of the soil (Fecan et al., 1998), and surface vegetation (Pierre et al., 2012). Low soil moisture, minimal vegetation, and strong winds provide optimal conditions for dust lofting. Dust lofting is internally computed for 7 particle radius bins (0.15, 0.26, 0.47, 0.83, 1.50, 2.65, and 4.71 μm) (Tegen and Fung, 1994; Tegen and Lacis, 1996; Ginoux et al., 2001) which are then combined into 2 dust modes (sub-micron and super-micron) to minimize computational demands. Lofting is parameterized by dust particle size according to an inverse lofting relationship between aerosol size and threshold friction wind velocity (Alfaro and Gomes, 2001; Shao, 2001). Dust lofting is therefore ultimately computed in terms of dust mass flux as a function of particle size and the parameters discussed above, as well as the surface erodible fraction (Ginoux et al., 2001; Saleeby and van den Heever, 2013). The RAMS dust lofting parameterization is, thus, based on the Global Ozone Chemistry Aerosol Radiation and Transport Model (GOCART), with modifications to include combined dust size bins, soil clay fraction effects, vegetation influences, and variable erodible fraction specifications.

## 2.2 WRF-Chem Specifications

The WRF-Chem model version 3 (Skamarock et al., 2008; Grell et al., 2005; Fast et al., 2006), hereafter referred to just as "WRF", was also used in this study to simulate the same dust lofting event over the Arabian Peninsula. WRF was run in a one-way nested grid configuration from 2-5 August 2016 with the outer Grid-1 at 15 km horizontal spacing and inner Grid-2 at 3.0 km spacing and 50 hybrid sigma-pressure levels. The WRF model domains cover nearly the same geographical area as

the RAMS simulation domain shown in Fig. 1. WRF was run with the GOCART dust aerosol module (Ginoux et al., 2001), Morrison two-moment microphysics (Morrison et al., 2005,2009), specified aerosol optical properties (Barnard et al., 2010), RRTMG longwave radiation (Iacano et al., 2008), Goddard shortwave radiation (Chou and Suarez, 1999), NOAH land surface model (Niu et al., 2011; Yang et al., 2011), BMJ cumulus parameterization on the coarse grid domain (Janjic, 1994),

and MYNN level 3 boundary layer parameterization (Nakanishi and Niino, 2006,2009). A summary of the WRF model configuration and a general description of the physics packages used for simulations in this study are given in Table 2.

**2.3 Dust Erodible Fraction Experiments**

This study uses the same relative dust lofting physics for each test simulation in both RAMS and WRF. The dust lofting in

both models largely follow the GOCART methods from Ginoux et al. (2001). However, WRF retains dust in all lofted size bins while RAMS combines these into two bins, as mentioned previously.

In the RAMS and WRF experiments, only the surface soil erodible fraction, which ranges from 0.0 to 1.0 (0 % to 100 % erodible), will be varied in the three methods that are now described. These erodible fractions are shown in Figure 2. The (1)

"Idealized" method is similar to that used by Seigel and van den Heever (2012) in which an erodible fraction of 1.0 was used over their limited area model domain when simulating dust lofted by strong convection. Their study indicated lofted dust concentrations similar to those reported in severe dust storms over the southwest United States. The (2) "Ginoux" method uses the 1.0-degree global dataset of erodible fraction associated with Ginoux et al. (2001), which is shown in Fig. 2a, mapped to the Arabian Peninsula domains. The (3) "Walker" method uses a high resolution (~1.0 km) dataset of erodible

fraction (Walker et al., 2009), which is shown in Fig. 2b mapped to the Arabian Peninsula domains. These three methods of specifying erodible fraction are described in further detail below.

The Idealized method represents the upper bound on potential dust lofting given that any grid cell in the domain with conditions that are favorable for dust lofting (strong wind, dry soil, bare soil, and favorable soil type) can indeed loft dust

with an erodible fraction of 1.0. We suspect that the Idealized method, while useful for idealized simulations (e.g. Seigel and van den Heever, 2012), will produce extreme dust lofting in case study type simulations as are performed herein. However, it is useful to examine the upper bound of lofted dust that could be expected within a given case study and modeling framework. The Walker database erodible dust locations were identified manually using satellite data, and thus, this database identifies specific locations at approximately 1km resolution where dust is known to be available for lofting. Known dust

locations in the Walker database are designated with an erodible fraction of 1.0. The Ginoux database identifies more expansive dust lofting areas, compared to the Walker database, but with lower erodible fractions. The Ginoux database is based on the fraction of erodible sediment associated with topographic depressions as used in the GOCART model. The analysis that follows will refer to these three varying methods of assigning dust erodible fraction using the terms "Idealized", "Ginoux" and "Walker", and it will compare the varying amounts and locations of dust that is lofted according to these

different specifications of erodible fraction. Further, simulations were also run without dust lofting, denoted as "No-Dust", to provide a baseline comparison against simulations lofting dust. A summary of these simulations is provided in Table 3.

While sea salt was generated in these simulations and initial pollution aerosols are present in RAMS, and while WRF continues to generate a variety of aerosol species, the amounts of these aerosols are relatively small compared to the amounts of dust generated in all of these case study simulations. As such, this analysis will focus largely on the dust aerosols, with specific emphasis on the varying amounts of dust emitted to the atmosphere and its subsequent influence on the radiation and surface energy budget and temperature profile as a result of the dust erodible fractions being utilized.

It is important to note that the goal of the current analysis is to determine the sensitivity of the radiation budget and thermal response, within each modeling framework (RAMS and WRF), to the presence of lofted dust that varies with the geographical specification of dust erodible fraction. It is not the intent of this study to examine and interpret the modeling differences arising due to the model frameworks being utilized (i.e., RAMS vs. WRF). As such, differences in the model setup and parameterizations (with the exception of the dust lofting parameterizations) are not being considered within the

scope of this investigation.

### 2.4 Case Study Description

The dust lofting event simulated herein occurred from 2-5 August 2016 over the Arabian Peninsula with primary lofting occurring from (1) northerly flow over central Saudi Arabia and (2) southerly flow from coastal Oman to the United Arab

Emirates (UAE) (Figure 3). For the duration of this event, there is large scale high pressure and anti-cyclonic flow aloft over the Arabian Peninsula as inferred from the GDAS-FNL 500mb heights on 0000 UTC 4 August 2016 (Fig. 3a). This analysis time is 48 hours into the RAMS and WRF simulation times on the parent Grid-1, 24 hours into the simulations on the high-resolution Grid-2, and is just prior to the model analyses of instantaneous vertical profiles of radiative and thermal fields to be discussed in the sections that follow.

The GDAS-FNL streamlines at 925 mb display the near surface flow that impacts dust lofting (Fig. 3b). The 00Z August 4, 2016 streamline analysis exhibits cyclonic flow over the southern Persian Gulf. There is northerly flow over central Saudi Arabia that leads to a large dust source in that region being transported southward as seen in the Moderate-resolution Imaging Spectro-radiometer (MODIS) Aqua satellite image (Figure 4a). The southerly to southwesterly flow over Oman and

the eastern Rub al Khali mobilizes regional dust sources. Dust lofted in this area is transported to the northeast toward the UAE where additional mobilized dust is added from local sources. The wind field then transports the lofted dust in the southeasterly flow over the Persian Gulf where the mineral aerosols become quite evident as a highly visible plume over the Persian Gulf by 0930 UTC Aug 4 (Fig. 4). Due to the similar coloration between the dust mass and land mass in the visible imagery, the dust plumes over the Arabian land area are difficult to distinguish from the background. MODIS retrieved AOD

associated with the lofted dust is shown in figure 7 for two satellite overpasses during this event. The retrievals reveal substantial amounts of dust associated with plumes over both Saudi Arabia and the Persian Gulf. The MODIS AOD retrievals are discussed in further detail in section 3.1.

Conditions are very warm over the land regions at this time, as seen in the GDAS-FNL 1000mb temperature field (Fig. 3c) with temperatures exceeding 44° C in some locations. The equivalent RAMS and WRF simulated fields of 500 mb heights, 925 mb streamlines, and 1000 mb temperature from the Walker simulation on Grid-1 are shown in Figs. 3d-f and Figs. 3g-i, respectively. The modeled geopotential height fields both indicate broad scale high pressure similar to the GDAS data. The modeled streamlines also depict the strong northerly flow over Saudi Arabia and onshore southerly flow over Oman and

Yemen associated with the two main dust plumes.  The model simulated 1000 mb temperatures tend to be 2-4° C higher over the Saudi interior in both models. Differences in topography and land-surface parameterizations may account for the discrepancies in the magnitude of predicted temperature between the reanalysis and the RAMS and WRF simulations, while the differences in fine scale horizontal variability are likely the result of differences in grid resolution between the reanalysis and model simulations. Though not shown, near-surface humidity is low and soils are very dry and conducive to dust lofting,

with mean volumetric soil moisture values between 0.04-0.05 $m^3$ $m^{-3}$.

### 3 Dust Lofting and AOD Comparisons

To obtain a first glimpse of the differences in the amount of dust generated via the various dust lofting erodible fraction assumptions, the total dust mass at the lowest model level (36 m) in RAMS and near the surface (945 hPa) in WRF at 0930

UTC 4 Aug is shown in Figure 5. This is the same time as the satellite image in Fig. 4. It can be seen that the Idealized method in RAMS (Fig. 5a), used by Seigel and van den Heever (2012), produces an extraordinary amount of dust compared to the simulations using constrained lofting locations and erodible fraction. The Idealized method appeared to perform well for the short-term, idealized Seigel and van den Heever (2012) study, but it does not appear to produce realistic results for a lengthier simulated case study environment. The Idealized WRF simulation also produces very unrealistically high values of

dust in many areas of the domain (Fig. 5d). As mentioned earlier, the Idealized simulations are representative of an upper bound on lofted dust that can be generated within a given modeling framework. As such, the thermodynamic and radiative effects of dust, to be discussed, in the Idealized dust lofting scenario represent an upper bound on the dust-related feedbacks.

The Ginoux and Walker database applications are surprisingly similar in their dust amounts and locations in RAMS (Fig.

5b,c) and in WRF (Fig. 5e,f). In both models, the Walker simulation produces more fine-scale spatial variability with respect to the lofting locations due to the precise, high resolution nature of the database. It also leads to greater amounts of near-surface dust mass in some locations, such as central Saudi Arabia, southeast Oman and northwest Oman, since the erodible fractions in these areas are not as diffuse as in the Ginoux data. It should be noted that while the Walker database leads to greater spatial variability in simulated lofted dust, this does not imply that the Walker simulations are more accurate than the

Ginoux simulations with respect to the net amount of lofted dust across the model domains. Walker et al. (2009) provide a quantitative assessment of the use of high-resolution point source dust locations.

In sections 3.1 and 3.2 that follow, we present AOD estimates from two MODIS overpasses and two AERONET stations that were available within the domain during this dust event. Given the sparsity of the MODIS and AERONET data available during this event and the limited coverage over the model domain, the comparisons with the simulated AOD are made in a qualitative manner. Our intent is to broadly demonstrate that the RAMS Ginoux and Walker simulations were able to generate the Saudi and UAE dust plumes at the approximate location and of similar AOD compared to the limited dust observations. From there, we focus on examining the potential range of radiative effects of the Saudi dust plume simulated with varying specifications of dust erodible fraction which lead to varying amounts of lofted dust.

## 3.1 MODIS vs. MODEL AOD

The RAMS total aerosol optical depth (AOD) at 550 nm wavelength was computed offline via RAMS aerosol output. This output is, as expected, highly dominated by the dust modes. Aerosol particles were first grown hygroscopically to equilibrium with model relative humidity in each grid box using κ-Köhler theory (Petters and Kreidenweis, 2007). Aerosol extinction, and thus, AOD, is a function of the real part of the index of refraction. A representative real refractive index for dust of 1.53 at 550 nm was assigned based on surface observations (such as AERONET) (Dubovik et al., 2006; Giles et al., 2012), radiative closure studies (Wang et al., 2003; Christopher and Wang, 2003), and laboratory studies (Di Biagio et al. 2019). This value matches that used in the RAMS parameterization of dust radiative effects. Refractive indices for hygroscopic species were adjusted based on volume mixing with water. Representative extinction coefficients for each model grid box were then calculated for each aerosol species using Mie theory (Bohren and Huffman, 1983). AOD in each 2-D model column was then calculated for each species using the extinction coefficients and heights in each column grid box and then summed for all aerosol species to produce an estimated total AOD. The WRF AOD is computed via Mie theory during runtime and is output as a standard 2D quantity. Similar to RAMS, a real refractive index of 1.53 for dust was used in WRF for generating AOD at 550 nm wavelength.

The 550nm AOD at 0930 UTC 4 Aug for each of the RAMS and WRF test simulations is shown in Figure 6. The figure panels coincide with the same panels of dust concentration from Fig. 5 for the same time. Maxima in AOD tend to coordinate with the maxima in near-surface dust concentration. AOD for the Idealized case is unrealistically high in both RAMS and WRF as expected from the extreme near-surface concentrations of dust (Fig. 5a,d). The RAMS Ginoux and Walker dust simulations indicate dust plume AOD values in the 1.5-2.5 range associated with the UAE and central Saudi dust plumes. The RAMS Ginoux simulation generates a plume over the UAE and Persian Gulf that is more expansive than that in the Walker simulation, though maximum AOD values are less than in the Walker generated plume. This is perhaps not unexpected given that the Ginoux dust sources cover a relatively larger area but with lower erodible fraction. The Saudi

dust plume in the Walker simulation is both more expansive and contains higher maximum AOD at the time shown. This could also be expected given the relatively high density of dust sources in the Walker database in this area and the relatively low erodible fraction in the Ginoux database over central Saudi Arabia (see figure 2). The Ginoux and Walker simulations from WRF also show the relatively highest AODs associated with the Saudi dust plume and the UAE dust plume. However, dust plume AOD values from WRF (0.5-1.0) tend to be noticeably lower than those from RAMS (1.5-2.5) at the time shown. The lower AOD in WRF compared to RAMS results from much less generation of lofted dust in WRF (Fig. 5).

The MODIS Aqua and Terra 550 nm AOD at 10 km resolution (mapped to the RAMS 2 km domain) at approximately the same time are shown in Figure 7a,b. These MODIS-based AOD retrievals are obtained from MODIS Collection 6.1 product and are further processed with retrievals over the coastal turbid water (Wang et al., 2017); they have an uncertainty of ~15-20 % over land and 10 % over ocean (Levy et al., 2013; Hsu et al., 2013; Wang et al., 2017) with potential reduction of accuracy at high AOD (Levy et al., 2013). The MODIS data shown represent the two overpasses available during this case that have the least amount of missing data from the retrieval.

Similar to the RAMS Ginoux and Walker simulations, the MODIS AOD values in the dust plumes over the UAE, Oman, and Saudi Arabia are also in the 1.5-2.5 range with some pixels perhaps indicating even higher values over the Persian Gulf. The RAMS Ginoux and Walker simulations are, thus, performing favorably with respect to generating amounts of lofted dust that lead to AODs that are similar in magnitude to the limited remote sensing observations of the two main dust plumes. A visual comparison of the dust plumes among the MODIS visible image, MODIS AOD retrievals, and model AOD indicate that the RAMS simulated plumes are slightly displaced, with the Saudi plume located slightly north of the observed location and the UAE plume not extending as far north into the Persian Gulf as that observed. We also note that these simulations generate a more distinct gap of lower AOD between the two plumes. While variability in the transport of the dust plumes in the RAMS simulations leads to some discrepancy in the plume placement, both the RAMS Ginoux and Walker simulations produce dust plumes that are similar in expanse and AOD to those shown in the MODIS AOD. As such, an investigation of the impacts of locally lofted dust in these simulations may offer insight into the potential radiative and thermal response across a range of realistically simulated dust plumes that vary due to differences in dust erodible fraction.

Both RAMS and WRF generate the two key dust plumes from surface lofting over central Saudi Arabia, Oman and the UAE, with RAMS producing dust plume AODs in the Ginoux and Walker simulations that reflect the AOD values from the limited observations. Results from the Idealized simulations from both models indicate the need for application of dust source databases to dust lofting schemes, while simultaneously demonstrating the anticipated upper range of potential dust lofting within each given model framework. For the sake of brevity and given that WRF tends to under-predict dust plume AOD in the Ginoux and Walker simulations, the remainder of this manuscript will now focus only on results from the RAMS simulations. A more extensive model inter-comparison needed to understand the differences in dust mass and AOD

between RAMS and WRF is left for future investigation. Given that both models use the lofting techniques of GOCART and the same erodible fraction databases, we speculate that the prediction of the near-surface wind speed, soil moisture, dust deposition, and dust binning techniques may all play a role in explaining the difference in amounts of simulated lofted dust.

## 3.2 AERONET vs. RAMS AOD

The time series of AOD (at 500 nm) from the Aerosol Robotic Network (AERONET) (Holben et al., 1998,2001; Smirnov et al., 2002) from the Mezaira, UAE site (level 1.5 data) (Fig. 7c) and Kuwait University, Kuwait site (level 1.0 data) (Fig. 7d) are shown along with the associated RAMS total AOD (at 550nm) time series for the corresponding grid point locations. The locations of these two sites are indicated by the large black dots in Fig. 7a,b. AERONET data are only available during daylight hours. At the Mezaira site, the simulated AOD for the Mezaira grid point location most closely agrees with the AERONET AOD in the Ginoux and Walker simulations. The No-Dust simulation demonstrates the substantial contribution that dust, when included in the simulations, makes to the total AOD. The excessively high AOD in the Idealized simulation indicates that constraints on the erodible fraction are necessary to generate a reasonable prediction of both dust mass and AOD. Both the Ginoux and Walker simulations appear to underestimate the AERONET AOD at the Mezaira grid point. However, both simulations show an increase in AOD from Aug 3 to Aug 4 as in seen in the observations. As noted earlier, the RAMS simulated dust plumes are slightly displaced compared to the plumes seen in the MODIS data. Caution should be exercised when making single grid point comparisons like these as they can be deceiving when key model features are shifted within the simulations. The Walker AOD time series at Mezaira remains relatively low since the dense part of the plume is shifted a bit to the east and the Mezaira grid point in the model sits within the gap region between the plumes. The Ginoux dust plume is broader than the Walker plume due to the widespread nature of the dust source locations in the region as discussed earlier. As such, the Ginoux AOD time series displays higher AOD than the Walker simulation, and more closely compares to the AERONET AOD at Mezaira during the dust plume passage.

To demonstrate the range of spatial variability in simulated AOD and the need to consider plume displacement, the time series of AOD in the Ginoux and Walker simulations for the grid point 2-degrees to the east of Mezaira are shown as the colored, dotted lines in Fig. 7c. This location to the east of Mezaira is more clearly in the simulated dust plume, with the Ginoux simulation matching well with the Mezaira AERONET, while the Walker simulation produces higher AOD that perhaps represents the maximum aerosol concentrations within this plume. As such, where the plume is sampled and where the model places the plume in the simulations greatly impacts the comparison to individual grid point observations.

The MODIS AOD values are also interpolated to the Mezaira AERONET location as shown by the large blue and orange dots on figure 7c. For the two given overpasses the MODIS grid point estimates are lower than the AERONET AOD. While part of the difference could be attributed to uncertainty in AOD from MODIS and AERONET data for high AOD conditions, the horizontal interpolation of the MODIS pixels to the Mezaira point location occurs in an area with gradients in AOD and

near missing data pixels. As such, the interpolations likely reduce the MODIS AOD grid point estimates at Mezaira. However, both the MODIS and AERONET AOD data indicate the presence of a substantial dust plume that is well above the background state seen at the beginning of the time series in figure 7a.

The northern Kuwait University AERONET site is well removed from the intense dust episodes and is located in an area with only small horizontal AOD gradients indicated in the model output. The AOD values vary from ~0.2-0.7 on Aug 3 and from ~0.3-0.5 on Aug 4. The Ginoux and Walker simulations indicate very similar AOD predictions with values from 0.2-0.7 during this two-day time frame. This AERONET site is clearly not being impacted by dust plumes during this time. Kokkalis et al. (2018) examined a decade of AERONET data at this Kuwait City location and found mean daily AOD values

of 0.45±0.29, which is similar to the AERONET and RAMS model AOD during this event (Fig 7d). They also identify dust storm mean AODs as having values of 1.04±0.32 with some larger events exceeding an AOD of 1.5. This is rather representative of what is seen in the Mezaira AERONET and simulated UAE dust plumes. The largest contributor of dust storms to this location originate from Saudi Arabia. While the timing of the simulated AOD maxima and minima do vary slightly from the AERONET observations at the Kuwait site, the overall prediction is quite reasonable given the inherent

difficulty in accurately simulating all of the variables that feed into prognostic dust lofting and its transport and removal mechanisms.

It should be noted that the Mezaira site is located between the central Saudi Arabian dust maximum and the dust plume over Oman and eastern UAE, as can be seen in the satellite imagery and model output. Since the Ginoux and Walker simulations appear to offer a reasonable prognosis of dust lofting and transport, one might expect observed AOD values as high as 2.5-

3.0 in the nearby dust plume in the eastern UAE. Given the significant spatial and temporal variability in dust amounts, this is an area of the world that could significantly benefit from additional AERONET stations that could assist with model initialization and/or validation of major dust events. This would also help to determine if model simulations of AOD magnitudes are reasonable but could use improvement regarding the placement of specific dust storm events.

**4 Dust Impacts on Radiation**

For the remainder of the discussion, the analyses of the model output will focus on the direct radiative effects of dust in the RAMS model, how these effects vary among the three simulations with various dust erodible fractions, and how this ultimately impacts the temperature profiles over significantly dusty arid regions. To isolate the specific dust effects on

radiation over this simulated domain that includes flatlands, mountainous terrain, coastal zones, and ocean area, we examined an inland area over the central Saudi Peninsula where dust concentrations are rather high and cloud cover was minimal. This sample region was chosen so as to exclude from the analysis any potential variability in cloud cover and maritime influences among simulations. Time series and vertical profiles of several quantities will be presented as area averages within the 5°x5° box that is denoted in Fig. 1b. Instantaneous vertical profiles will be shown for nighttime at 0200

UTC (~0600 LST) and for daytime at 1000 UTC (~1400 LST) on the 4th of August 2016. These times were chosen since they represent the approximate times of the peaks in nighttime cooling and daytime heating, respectively.

## 4.1 Dust and Temperature Time Series

To begin, Figure 8a depicts the time series of horizontally averaged integrated dust mass over the analysis box region (Fig. 1b) from the high-resolution RAMS model domain. From this perspective, the Ginoux and Walker databases give very similar results, with the Walker database leading to slightly greater dust amounts, likely due to the localized dust lofting areas with high erodible fraction over central Saudi Arabia, southeast Oman and northwest Oman (see Figs. 2b and 5c). The Idealized dust simulation produces dramatically greater dust amounts that generally continue increasing over time as a result

of lofting rates exceeding deposition rates. Some of the radiation and temperature responses to the dust evolve over time as the model domain transitions from a cleaner to dustier environment. As such, the focus of the analysis will largely be on day-2 (0000 UTC Aug 4 to 0000 UTC Aug 5). However, some linkage can be made to the model state on day-1 (Aug 3), which is displayed in the time series for completeness. For the remainder of the manuscript, the Idealized dust case is periodically referred to as containing an "extreme" dust amount while the Ginoux and Walker cases will be referred to as those with

"moderate" dust amounts.

The time series of near-surface temperature (Fig. 8b) reveals a couple of key patterns that arise by day 2. The minimum nighttime near-surface temperature tends to increase with increasing dust amount while the maximum daytime temperature is lowest for the extreme dust amounts in the Idealized case and highest for the moderate dust amounts in the Ginoux and

Walker cases. The greater near-surface daytime temperature in the moderate dust cases is only slightly greater than the No-Dust control case, while the temperature reduction of ~3° C in the Idealized case is comparatively large. In general, very large amounts of lofted dust tend to reduce the overall near-surface diurnal temperature range. Moderate amounts tend to have variable impacts on the diurnal near-surface temperature cycle. The analyses that follow will attempt to identify which radiative components impact these noticeable changes in nighttime and daytime temperature extremes.

## 4.2 Daytime Radiative Fluxes

This section examines daytime (1000 UTC, ~1400 LST for 4 August) vertical profiles of quantities averaged over the analysis box region discussed earlier. This daytime snapshot is taken around the time of maximum: (1) downward shortwave radiation (Fig. 8c), (2) upward longwave radiation (Fig. 8d), (3) surface sensible and latent heat fluxes (Fig. 8e,f), and (4)

near-surface temperature (Fig. 8b). There is some time lag between maxima in radiation, surface fluxes, and temperature, but they tend to occur within about an hour of one another in this case. Vertical profiles of dust concentration in figure 9a indicate the lowest dust concentrations from the Ginoux simulation, slightly greater values from the Walker simulation and substantially high concentrations for the Idealized simulation. While the Ginoux and Walker simulations have very similar amounts of lofted dust, the Walker simulation likely produces greater amounts of dust due to a relatively large number of

source locations over the analysis region and comparatively low erodible fractions in the region from the Ginoux database (see figure 2).

Of the radiative quantities examined, the daytime surface downward shortwave radiation (Figure 9f) is the radiative flux that is most greatly impacted by dust, which is in keeping with previous findings (e.g. Slingo et al., 2006; Marsham et al. 2016). Scattering and absorption of dust at solar wavelengths tends to reduce the amount of shortwave radiation penetrating downward through the atmosphere. This impact is found to be greatest near the surface, which corresponds to increasing dust amounts near the ground (Fig. 9a). The downward shortwave profile shows reductions from 200-800 W m$^{-2}$ that scales with the trend in average dust amounts. The shortwave reductions by dust of ~200-250 W m$^{-2}$ for the Ginoux and Walker simulations are similar to those shown by Slingo et al. (2006) and Kosmopoulos et al. (2017) for corresponding dust AOD in the range of 1.5-2.5. These reductions by themselves would tend to induce a cooling effect near the ground during the daytime by limiting the land surface heating. This effect is manifest in Fig. 8, which shows that the daytime surface sensible (Fig. 8e) and latent heat (Fig. 8f) fluxes are also reduced under conditions of greater dust loading. The daytime surface upward longwave radiation (Fig. 8d) is also reduced with increasing dust and decreasing surface insolation since the ground is heated less effectively and emission temperatures are lower. Again, the most noticeable impacts are on day-2 in association with the greater lofted dust amounts (Fig. 8a).

The ultimate impact of dust on the temperature profile is determined by a complex balance between upward and downward short and longwave radiative fluxes (Fig. 9), as well as ground heat storage and surface sensible (Fig. 8e) and latent (Fig. 8f) heat fluxes that regulate the boundary layer processes and, thereby, the dust vertical profiles in the boundary layer. Marsham et al. (2016) also highlight the complex balance of radiative fluxes in determining the net radiative response to dust loading. The ability of dust to scatter and absorb shortwave radiation (e.g. Banks et al. 2014; Marsham et al. 2016) and absorb and emit longwave radiation (e.g. Haywood et al. 2005; Marsham et al. 2016) influences these fluxes, thus impacting the temperature profile. While the downward shortwave response to dust is rather straightforward, the other fluxes are more variable and trends are less monotonic. The upward shortwave (Fig. 9g) response to dust follows the downward shortwave trend since surface albedos are similar among experiments. The downward longwave flux (Fig. 9d) in the lowest 3km, where most of the dust resides, corresponds to downward re-emission of radiation absorbed by dust. As such, this shows a trend of greater downward longwave flux with an increase in dust mass in the lowest several kilometers. The upward longwave flux (Fig. 9e) displays only small differences near the ground for the moderate dust amounts but decreases under extreme dust loading. The noticeable decrease in upward longwave flux in the extreme dust lofting case results from cooler surface temperatures (Fig. 8b and 9c), and thus, lower thermal emission rates. The increase in downward longwave emission in the dust layer, which partially offsets the cooling effect of reduced shortwave at the surface, has also been noted in observational studies (e.g. Slingo et al., 2006; Hansell et al., 2010; Marsham et al., 2016). Generally, the changes in fluxes from the No-Dust case increase with increasing dust amount from the Ginoux to Walker to Idealized dust simulations. However, the

differences between the Ginoux and Walker simulated fluxes are smaller compared to those from the Idealized simulation due to similar dust loading.

The total or net radiative flux (downward minus upward sum of shortwave and longwave fluxes) tells a more concise story of the atmospheric radiative impacts of dust (Fig. 9h). Near the surface there is a monotonic trend of decreasing total radiative flux that trends with increasing dust mass from the Ginoux to Walker to Idealized dust simulations. A decrease in surface net radiative flux with increase in dust loading was also found by Marsham et al. (2016). Similar to the RAMS simulation results, Marsham et al. (2016) revealed that the dust-induced reduction in surface shortwave heating is greater than the corresponding increase in longwave heating, with a resulting reduction in net radiative fluxes. By itself, the difference in the magnitudes of the total radiative flux among the various dust mass conditions suggests that, near the surface, the presence of dust should induce a cooling effect. Above the surface, the total flux increases with height, with the rate of increase (the slopes) being steeper for greater dust mass. The profiles increase within the dusty layers and then assume a neutral slope aloft that is similar to the No-Dust scenario. Above ~6 km AGL the comparative total radiative flux profiles show a monotonic increase with increasing low-level dust mass – a trend that is opposite to that of the near-surface. This behavior is due to the steepness of total flux profiles within the dust layers. A more positive value of total radiative flux corresponds to greater atmospheric accumulation of short and longwave radiation, and thus the potential for greater warming. The trend in total radiative flux above ~6 km suggests that low-level dust layers can induce a net warming effect in the column above them.

In addition to the magnitude of the total radiative flux, the radiative flux divergence or radiative heating rates, also contribute to atmospheric heating and cooling associated with dust layers. The slopes of the total radiative flux profiles are indicative of the magnitude of the radiative flux divergence. The slopes are steeper for greater dust mass, which indicates greater radiative flux divergence within the dusty layers and stronger radiative heating rates. The associated radiative heating rate profiles (Fig. 9b) indicate a strong atmospheric radiative heating impact of dust from ~7 km to the surface that increases monotonically with increasing dust mass from the Ginoux to Walker to Idealized dust simulations. Observations have also shown increases in radiative heating rates with dust loading associated with increases in radiative flux divergence within the dust layers (e.g. Hansell et al., 2010; Marsham et al., 2016). The cooling effect of reduced surface net radiative fluxes can be countered by an increase in radiative heating within dust layers.

The resulting atmospheric temperature profile is thus controlled by a complex interaction among the: (1) magnitude of total radiative fluxes, (2) radiative flux divergence / radiative heating rates, (3) surface latent and sensible heat fluxes, and (4) atmospheric mixing. The area-mean low-level maximum daytime temperature profile (Fig. 9c) indicates a dust induced heating effect above ~600 m AGL, below which, there is a cooling effect imposed by extreme dust loading. In the lowest 200m or so, the extreme dust loading has reduced the daytime mean maximum temperature by about 2.5° C. The moderate

dust amounts lead to a small warming of about 0.3°-0.4° C near the ground despite reduced total radiation and reduced sensible and latent heat fluxes. This warming, which is only slightly more in the Walker simulation compared to the Ginoux simulation, appears to be induced by the increased net radiative heating rates shown in (Fig. 9b). Thus, except for the case of extreme dust loading, lofted dust tends to induce a net warming effect at the surface despite reductions to insolation. We

suspect that in our simulations, the weighting of dust toward the surface prevents substantial surface cooling except in the presence of very high and unrealistic dust loading. The in-layer atmospheric radiative heating rates counter balance the surface cooling effect of dust. Had the dust been concentrated in an elevated layer, we may expect to see a stronger and more consistent surface cooling during the daytime with the radiative heating rates concentrated in the dust layer aloft (e.g. Lau and Kim, 2007; Shell et al. 2007).

**4.3 Nighttime Radiative Fluxes**

This section examines nighttime (0200 UTC, ~0600 LST for 4 August) vertical profiles of quantities averaged over the analysis box region discussed earlier. At nighttime, the radiation and associated temperature responses behave differently compared to the daytime. Over the nighttime hours, the solar component of radiation is no longer a factor (Fig. 8c). The

surface upward longwave radiation trends (Fig. 8d), however, indicate a clear maximum increase of nearly 50 W m$^{-2}$ between the no-dust and extreme dust case in the pre-dawn hours of 4 August. Further, the overnight near-surface temperatures (Fig. 8b) are monotonically warmer in the dusty cases with a maximum difference just before dawn. The latent and sensible heat fluxes are relatively small at night (Fig. 8e.f), although there is a modest increase in the nighttime latent heat flux with dust loading that could be contributing to warmer near-surface temperature. Though not shown, this stronger

latent heat flux under warmer, dustier conditions may result from modestly stronger winds that occur in association with warmer temperatures and more boundary layer mixing.

The near-surface temperature and upward longwave radiation discussed above are not quantities that are independent of one another, but rather, one largely determines the other. As such, a closer examination of the nighttime radiation vertical

profiles in the hour just before dawn demonstrates the key controlling factors that impact the nighttime temperature response to dust loading.

First, the nighttime vertical dust concentration profile (Figure 10a) is very similar to the daytime profile with dust concentrations increasing from the Ginoux to Walker to Idealized simulations (Fig. 9a). The nighttime temperature profile

responds variably to dust loading between the near-surface layer and layers aloft in the moderate dust cases, but it is consistently warmer from the surface upward in the extreme dust case (Fig. 10c). Right near the surface, there is a monotonic increase in temperature with dust loading. The extreme dust event shows a temperature increase of ~3° C compared to the No-dust case while the moderate dust events indicate increases of ~1° C for the Walker simulation and ~0.5° C for the Ginoux simulation. The increase in near-surface temperature occurs in the moderate dust cases despite stronger radiative

cooling rates (Fig. 10b) in those cases compared to no-dust. This results from a comparatively smaller total radiative flux (less negative) near the ground (Fig. 10f), which implies reduced longwave emission. The slightly greater surface latent heat fluxes in the dusty simulations also enhance near-surface warming. As such, near the surface, the increase in latent heat flux and reduction in net thermal emission appear to play a greater role towards a warming effect of dust than the opposing

stronger radiative cooling rates in determining the nighttime near-surface temperature warming trend with dust loading.

Above the first few hundred meters, however, the temperature trend is non-monotonic and the moderate dust cases indicate a minor cooling impact of ~1° C. The downward longwave flux profile (Fig. 10d) shows a monotonic trend that increases with dust loading below ~3 km AGL with a maximum increase of over 100 W m$^{-2}$; this increase is associated with the dust layer

absorbing thermal radiation and re-emitting this back towards the ground (e.g. Slingo et al., 2006; Marsham et al. 2016). The upward longwave (Fig. 10e) trend is monotonic right near the surface, similar to what is seen in the associated upward longwave time series (Fig. 8d). Above 3 km, however, the moderate dust cases display slightly greater upward longwave fluxes compared to No-Dust, while the extreme dust case is consistently less than the No-Dust and moderate dust cases. The non-monotonic trend in the upward longwave results from a competition between the upward thermal emission near the top

of main dust layer and the amount of thermal radiation which is absorbed by the dust. The reduction in upward longwave in the extreme dust case is rather large and results from dust absorption of thermal radiation and warming of the layer, which prevents less longwave escaping to space. This effect is less substantial in the moderate dust cases, thus leading to a non-monotonic response in upwelling longwave radiation to an increase in dust mass.

The total radiative flux (Fig. 10f) below 1 km is monotonically reduced (less negative) with increasing dust concentration from the Ginoux to Walker to Idealized simulation, which suggests a near-surface warming effect. Above ~1 km AGL, however, the total flux trend is non-monotonic. Moderate dust loading leads to greater total fluxes (more negative) compared to No-dust, while extreme dust amounts lead to a reduction in the total flux (less negative), which indicates less longwave radiation is escaping. The increase in total flux aloft for the moderate dust cases suggests a cooling effect; further, the

radiative heating rates (Fig. 10b) also demonstrate a greater cooling effect. The combination of these influences leads to the slightly cooler temperature (Fig. 10c) above 600m in the moderate dust cases compared to No-dust. The warmer temperature in the column for the extreme dust, compared to No-Dust, is a result of the greater surface latent heat flux and diminished total radiative fluxes that offset the stronger radiative cooling from longwave flux divergence.

In the absence of solar radiation, the nighttime total radiative flux values are small relative to the daytime radiative quantities (e.g. Hansell et al., 2010; Marsham et al., 2016). As such, small changes in those factors impacting the radiation budget can more easily impact the ultimate balance in the heating or cooling near the ground and aloft at night. The dust impact on the radiative balance, and thus, the temperature profile at night, is only straightforward when comparing the No-Dust to the extreme dust simulations. More moderate dust events as in the Ginoux and Walker simulations do not produce consistent

nighttime monotonic trends in the radiation fluxes with height. As discussed earlier, the low-level atmospheric temperature response to dust loading involves a complex interaction among the magnitudes of the total radiative fluxes, the ground surface heat fluxes, and the radiative heating/cooling rate (which is a function of the vertical attenuation rate or vertical flux divergence of the radiative fluxes). These controlling factors tend to have smaller magnitudes at night, thus making the net

effect more sensitive to changes in dust loading. These results suggest that large dust loadings are necessary to generate consistent nighttime trends in radiation and temperature profiles, particularly above the surface. Otherwise, dust adds to the spread of uncertainty in these trends.

**5 Summary and Conclusions**

In this study, the direct radiative impact of dust and the resulting impact on the daytime and nighttime temperature profiles over extreme arid regions were examined in numerical simulations of a dust lofting event over the Arabian Peninsula (4-6 August 2016) that made use of three spatially varying specifications of surface dust erodible fraction. A simulation with no dust (labeled "No-Dust") was compared to simulations that used dust erodible fractions that were: (1) idealized with an erodible fraction of 1.0 in all land grid cells (labeled "Idealized"), (2) specified by the 1-degree resolution dataset from

Ginoux et al. (2001) (labeled "Ginoux"), and (3) specified by the ~1km high-resolution data from Walker et al. (2009) (labeled "Walker"). Simulations were performed using both the RAMS and WRF-Chem models for comparison. The Idealized method of specifying erodible fraction has been shown to be useful in short term idealized type simulations (e.g. Seigel and van den Heever, 2012), but it likely represents the comparative upper bound of potential dust lofting and radiative responses in each respective model.

Both models revealed that Idealized dust lofting generates unrealistically high concentrations of dust mass and AOD, while the Ginoux and Walker simulations exhibited much more similarity. They also showed that use of the Ginoux and Walker dust erodible fraction databases reduces the amount of lofted dust compared to the Idealized method and brings the dust mass and AOD to within values closer to observations. RAMS simulations using the Ginoux and Walker databases

generated AODs that were similar to MODIS and AERONET observations, while WRF tended to underestimate the AODs. The use of the high-resolution erodible fraction database in the Walker simulations tended to produce more focused dust plumes with peak AODs that were higher than the those in the Ginoux simulations due to the identification of localized areas of high erodible potential. Meanwhile, the Ginoux simulations tended to produce more expansive plumes of moderately high AODs due to the more expansive area of moderate dust erodible fractions in the Ginoux database. However, the mean dust

profiles from the analysis region in the RAMS simulations revealed only modestly higher dust concentrations in the Walker simulations compared to the Ginoux simulation. For the sake of brevity, the radiative impacts of dust from these simulations were presented solely from the RAMS simulations.

Due to the great variability in dust impacts on radiation between daytime and nighttime, our analyses treated these portions of the diurnal cycle separately and focused around the time of daytime maximal heating and nighttime maximal cooling. At either time, the resulting low-level temperature profile results from a combination of competing influences that include the magnitudes of shortwave and longwave radiative fluxes, radiative heating/cooling rates determined from the radiative flux divergence, and surface heat fluxes.

During the daytime, enhanced dust concentrations associated with surface dust lofting tend to reduce insolation, total radiative fluxes and surface heat fluxes, all of which induce a cooling effect. However, the stratification of dust in the lower atmosphere leads to enhanced radiative heating rates within these levels through changes in the radiative flux divergence which counteract the cooling effects of reduced total radiative flux magnitudes. The net result is a modest column warming effect for conditions of moderate dust concentrations in the Ginoux and Walker simulations. Thus, it appears that moderate dust loading may invoke strong responses in the profiles of upward and downward shortwave and longwave radiation while inducing only a small warming effect in the lower atmosphere. The extreme concentrations of dust in the Idealized simulations, while unrealistically high, demonstrate that the near-surface atmosphere will be substantially cooled, coupled with substantial warming aloft. This cooling occurs at the surface in the extreme dust case due to the large reduction in insolation overwhelming the increase in radiative heating rate. In summary, moderate dust amounts in the Ginoux and Walker simulations with constrained dust lofting tend to warm the near-surface and regions aloft while extreme dust amounts in the Idealized lofting simulation with unconstrained dust lofting tend to cool the near-surface layer and warm the regions aloft. The warming aloft increased with increasing dust loading from the Ginoux to Walker to Idealized simulations.

At night, the absence of solar radiation leads to much smaller total radiative and surface flux magnitudes which makes the resulting temperature profile more sensitive to small changes in the upward and downward longwave fluxes that comprise the total radiative flux. Dust effects on the radiative heating rates and radiative fluxes are more complex and not necessarily monotonic, which complicates assessing their impacts. The effect of increasing dust from the Ginoux to Walker to Idealized scenarios at night generates monotonically reduced total radiative fluxes (less negative) and increased latent heat fluxes near the surface, which overwhelms the non-monotonic increase in radiative cooling and leads to a slight warming near the surface. The near-surface warming is modest for the moderate dust cases but is more substantial for the extreme dust simulation. Above the lowest several hundred meters, however, the dust impact on temperature becomes non-monotonic due to corresponding non-monotonic trends in the radiative flux profiles. Drawing general conclusions of the impacts of dust on nocturnal temperature profiles is therefore difficult, since small fluctuations in the radiation streams and cooling rates can alter the signs of net heating/cooling. However, for moderate dust amounts in the Ginoux and Walker simulations, the above-surface temperature profiles promote slight cooling, while extreme dust loading promotes warming. In summary, increasing dust at night warms the atmosphere close to the surface but has variable effects above the surface layer depending on the dust amount; an extreme amount of dust tends to warm the surface and regions aloft.

The dust lofting simulations using databases of dust erodible fraction helped constrain the amount of dust lofting, thus, producing dust AODs that were comparable to the observed AODs in the associated dust plumes. In the mean radiative analysis of the Arabian dust plume over land, the presence of the simulated dust imposed substantial impacts on the individual shortwave and longwave radiative fluxes. However, shortwave and longwave fluxes tended to partially offset one another, and since the Arabian dust plume was weighted toward the surface, the in-plume radiative heating rates computed from the radiative flux divergence tended to compensate for the changes in the radiative fluxes. So, while the radiative impacts were substantial, the impact of the surface-based dust layer on the temperature profile in the lowest 2 km was only ~1° C or less.

Fewer dust-lofting modeling studies have focused on the Arabian Peninsula compared to North Africa and East Asia largely due to limited in situ observations and field campaign data for comparison (Prakash et al. 2015). However, the Arabian Peninsula is a substantial contributor to atmospheric dust loading, with annual dust emissions comparable to East Asia (Tanaka and Chiba, 2006). Most studies agree that the shortwave attenuation by dust aerosols tends to induce a surface cooling effect. However, as mentioned above, the vertical distribution of dust has a strong impact on the net surface radiative effects. Studies of dust associated with long-range oceanic transport, such as those originating from North African and East Asia (e.g. Tanaka and Chiba, 2006; Lau and Kim, 2007; Prakash et al. 2015), tend to examine the effects of dust suspended in elevated layers. When the dust remains linked to the surface and transport is short-range, as in this Arabian Peninsula event, the daytime surface radiative response is different from an elevated dust layer due to the competing effects of shortwave cooling and dust layer radiative heating. As such, there is a limited surface thermal response associated with moderate dust lofting in this study. Extreme dust lofting in this study was necessary to lead to substantial daytime surface cooling and nighttime warming. We speculate that had the Arabian dust layer been elevated, the surface daytime and nighttime temperature changes would have been more substantial.

The modification of surface heating and low-level temperature profiles by dust loading has important implications for the development and/or maintenance of mesoscale weather features that are generated in association with surface heating. Convection generated from heated thermals could be modified by daytime dust loading which could then in turn modify additional dust lofting associated with convective outflows. The thermodynamic impacts of dust loading could also impact sea breezes which are generated by differential daytime heating between ocean and adjacent land surfaces (Miller et al., 2003). Impacts on sea breezes could then impact local dust concentrations and spatial distributions since onshore sea breeze winds have the potential to loft and transport dust and concentrate dust along sea breeze fronts (Verma et al., 2006; Igel et al., 2018). The size-dependent dust emission schemes can be further improved by using the constraints from both shortwave and longwave satellite measurements (Xu et al., 2017). The potential radiative impacts of dust on mesoscale features associated with the littoral zone will be examined in greater detail in future work.

The results from the dust lofting simulations performed in this study emphasize the need for: 1) continued development of high-resolution dust erodible fraction databases across dust prone regions of the world, 2) further high-resolution numerical studies that can adequately resolve both location-specific dust lofting zones and mesoscale circulations that respond to dust-related effects, and 3) additional surface based observations of AOD (such as AERONET) in regions that are frequently impacted by dust lofting episodes. Advancement in each of these factors could lead to improvement in the ability to simulate the impacts of dust aerosols on radiation and mesoscale phenomena such as sea breezes. Given the expansive coastline of the Arabian Peninsula and the frequency and widespread occurrence of sea breezes in this region (Eager et al. 2008), improvements in prediction of dust lofting would be greatly beneficial.

## Author Contributions

The dust database of Walker et al. (2009) and case overview was provided by Annette Walker. The Walker dust database was translated for the RAMS model by Jeremy Solbrig. AOD products associated with RAMS aerosols were generated by Python-based AOD software courtesy of Samuel Atwood, Qijing Bian, and Sonia Kreidenweis. MODIS Aqua and Terra aerosol AOD retrievals were processed and provided by Yi Wang and Jun Wang. MODIS Aqua true color imagery and case study guidance was provided by Steven Miller. WRF model analysis was provided by Jennie Bukowski. RAMS model analysis and data investigation was provided by Stephen Saleeby and Susan van den Heever. The majority of the manuscript was written by Stephen Saleeby, with all authors providing manuscript input and edits.

## Data Availability

Model data and code for reproducing simulations are available upon request to the corresponding author.

## Competing Interests

The authors declare that they have no conflicts of interest.

## Acknowledgements

This work was supported by the Office of Naval Research under Grant # N00014-16-1-2040 with project titled, "*Advancing Littoral Zone Aerosol Prediction via Holistic Studies in Regime-Dependent Flows*". The simulation data are available upon request from the corresponding author, Stephen M. Saleeby. RAMS Initialization and nudging data provided by: National Centers for Environmental Prediction, National Weather Service, NOAA, U.S. Department of Commerce. 2000, updated daily. *NCEP FNL Operational Model Global Tropospheric Analyses, continuing from July 1999*. Research Data Archive at the National Center for Atmospheric Research, Computational and Informational Systems Laboratory. https://doi.org/10.5065/D6M043C6.

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

**Table 1. Summary of RAMS model grid setup and simulation configuration.**

| Model aspect | Setting |
| --- | --- |
| Grid | Arakawa C grid (Mesinger and Arakawa, 1976) |
| | 2 grids: Grid 1 used for forcing Grid2 |
| | Grid 1: 432x422pts, $\Delta x = \Delta y = 15$ km, $\Delta t = 30$ s |
| | Grid 2: 950x950pts, $\Delta x = \Delta y = 2$ km, $\Delta t = 6$ s |
| | 50 Vertical Levels |
| | $\Delta z = 75$m lowest level stretched to 750m aloft |
| | Model top at ~22 km AGL |
| Initialization | GDAS-FNL re-analysis 1-degree data |
| Boundary conditions | Lateral boundary nudging from gridded reanalysis (Davies, 1983) |
| Land-surface model | LEAF-3 (Walko et al., 2000) |
| Cumulus parameterization | Kain-Fritsch parameterization (Kain and Fritsch, 1993) on Grid-1 |
| Radiation scheme | Two-stream, hydrometeor-sensitive (Harrington, 1997) and aerosol-sensitive (Stokowski, 2005; Saleeby and van den Heever, 2013) |
| Turbulence scheme | Horizontal and vertical turbulent diffusion via Smagorinsky (1963) |
| Microphysics scheme | Two-moment bin-emulating bulk microphysics for eight hydrometeor species (Walko et al., 1995; Meyers et al., 1997; Saleeby and Cotton, 2004; Saleeby and van den Heever, 2013) |

**Table 2. Summary of WRF-Chem model grid setup and simulation configuration.**

| Model aspect | Setting |
| --- | --- |
| Grid | 2 grids: one-way nesting grid setup<br>Grid 1: 432x422pts, $\Delta x = \Delta y = 15$ km, $\Delta t = 60$ s<br>Grid 2: 950x950pts, $\Delta x = \Delta y = 3$ km, $\Delta t = 15$ s<br>50 Vertical Levels (hybrid sigma-pressure)<br>Model top at ~22 km AGL |
| Initialization | GDAS-FNL re-analysis 1-degree data |
| Boundary conditions | Lateral boundary nudging from gridded reanalysis (Davies, 1983) |
| Land-surface model | NOAH (Niu et al., 2011; Yang et al., 2011) |
| Cumulus parameterization | BMJ parameterization (Janjic, 1994) on Grid-1 |
| Radiation scheme | RRTMG (Iacano et al., 2008) with aerosol optical properties (Barnard et al., 2010) |
| Boundary Layer scheme | MYNN Level 3 (Nakanishi and Niino, 2006,2009) |
| Microphysics scheme | Two-moment Morrison (Morrison et al., 2005,2009) |
| Aerosol module | GOCART model (Ginoux et al., 2001) |

**Table 3. Summary of Simulations.**

| Simulation Name | Simulation Description |
|---|---|
| No Dust | No dust lofting permitted. |
| Idealized | Dust erodible fraction of 100% for grid cells with appropriate soil type and minimal vegetation. |
| Ginoux | Use of Ginoux et al. (2001) 1°x1° erodible fraction dataset mapped to model domain. |
| Walker | Use of Walker et al. (2009) ~1km dust erodible fraction dataset mapped to model domain. |

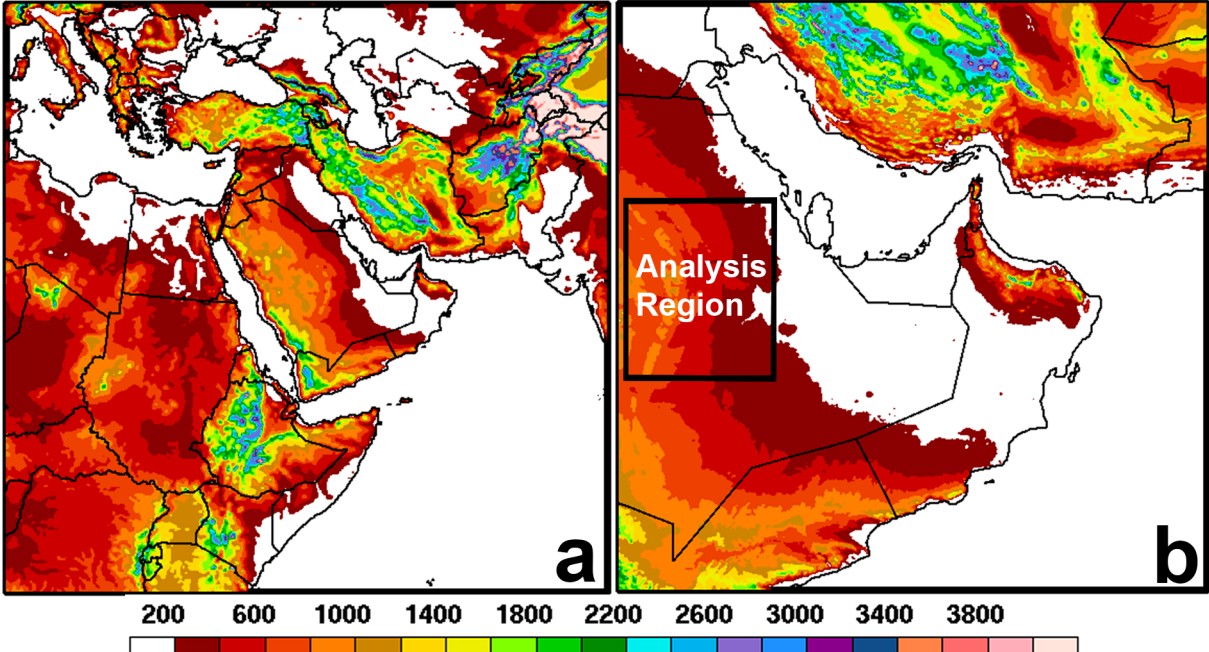

**Figure 1. RAMS simulation domains with topography (shaded) for the domains with grid spacings of (a) 15km (Grid-1) and (b) 2km (Grid-2). The inset box denotes the dusty inland analysis region for the area-averaged time series and vertical profiles that follow.**

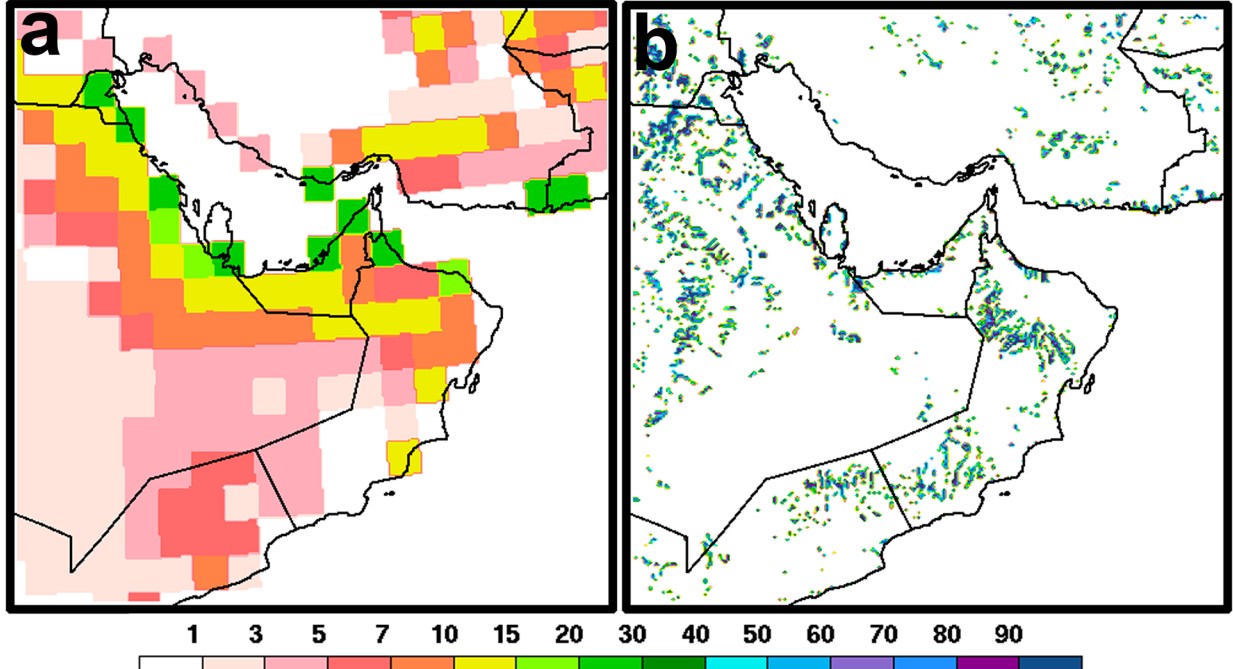

Figure 2. Erodible fraction potential for utilization in the dust lofting parameterization from (a) Ginoux et al. (2001) 1-degree dataset and (b) Walker et al. (2009) high resolution (1-km) database developed at the Naval Research Lab (NRL). Both datasets are shown mapped to the RAMS 2-km grid spacing domain. Values represent 0-100% erodible fraction.

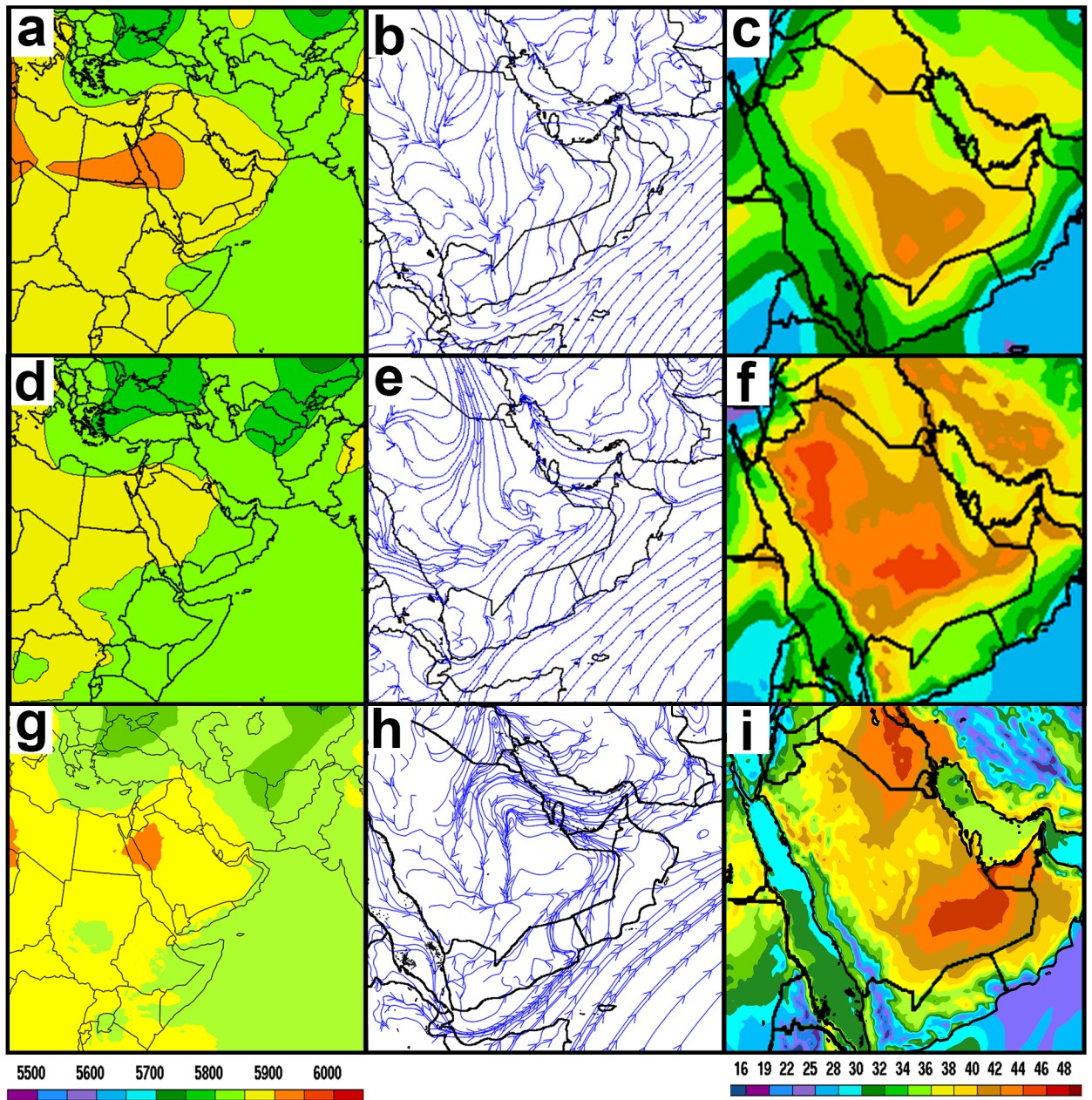

**Figure 3. August 4, 2016 at 0000 UTC: (a,d,g) 500mb geopotential height (m), (b,e,h) 925mb streamlines, and (c,f,i) 1000mb temperature (C) from (a-c) GDAS-FNL, (d-f) RAMS Grid-1, and (g-i) WRF-chem Grid-1.**

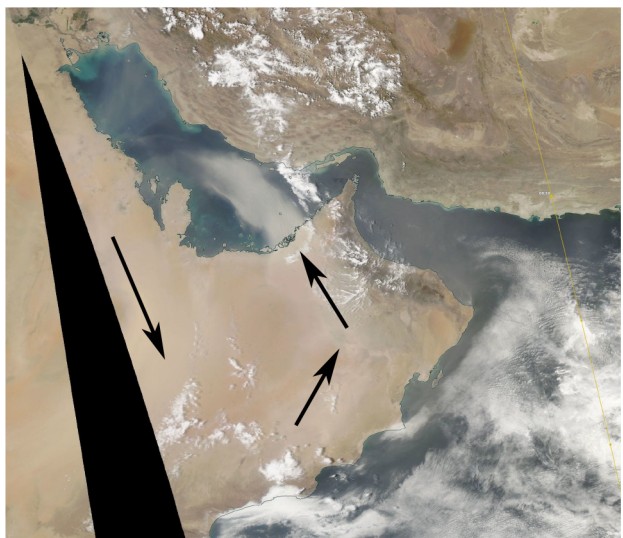

**Figure 4. Snapshot of MODIS visible satellite image from 0930 UTC 4 August 2016. Arrows indicate the main direction of lofted dust advection and transport.**

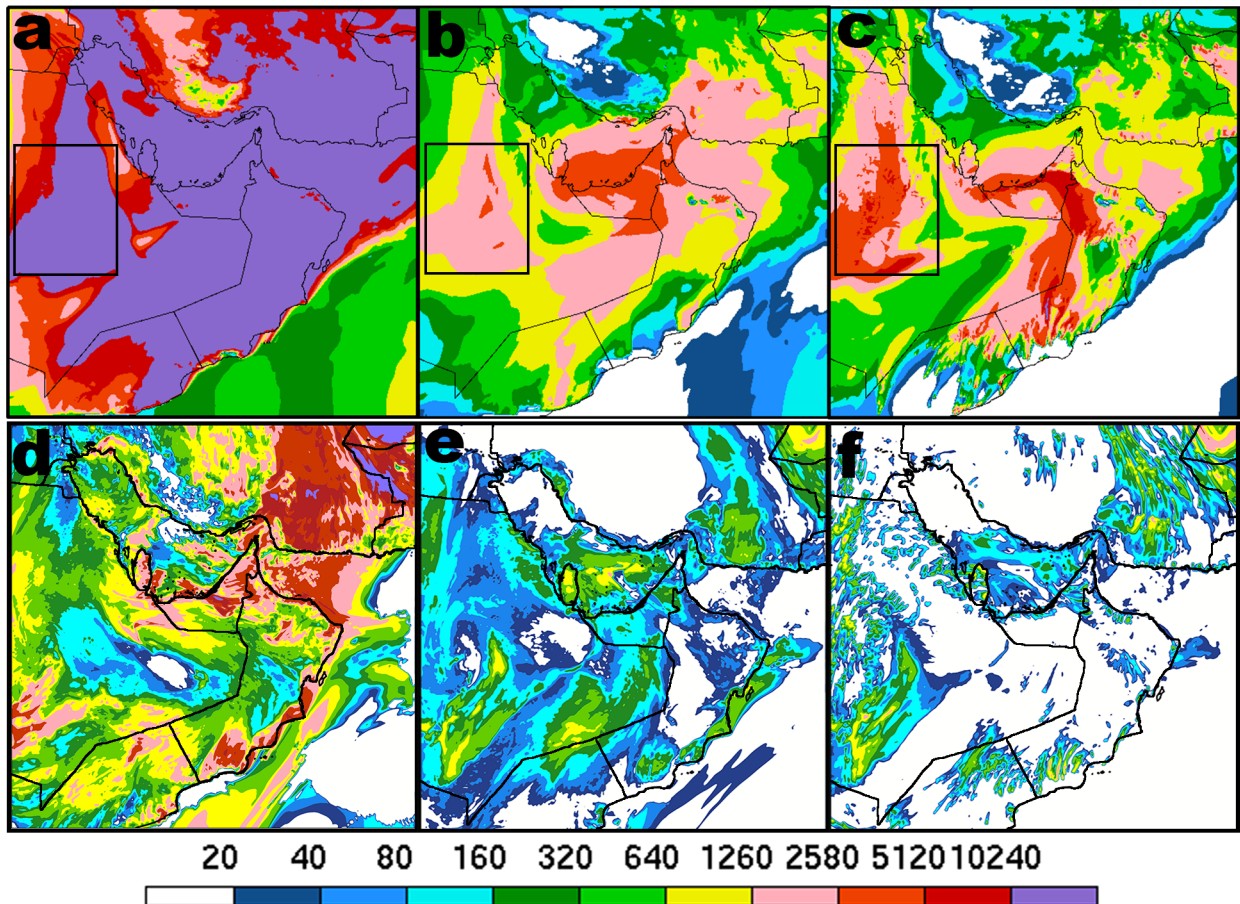

**Figure 5. Simulated dust mass (ug/m3) at 0930 UTC 4 Aug, 2016 from the lowest model level (36m AGL) in RAMS and from the near-surface (945hPa) in WRF-chem using the (a,d) Idealized dust lofting, (b,e) Ginoux dust sources, and (c,f) Walker dust sources from the simulations from (a-c) RAMS and (d-f) WRF-chem. The black box over the central Arabian Peninsula denotes the analysis region shown in Figure 1.**

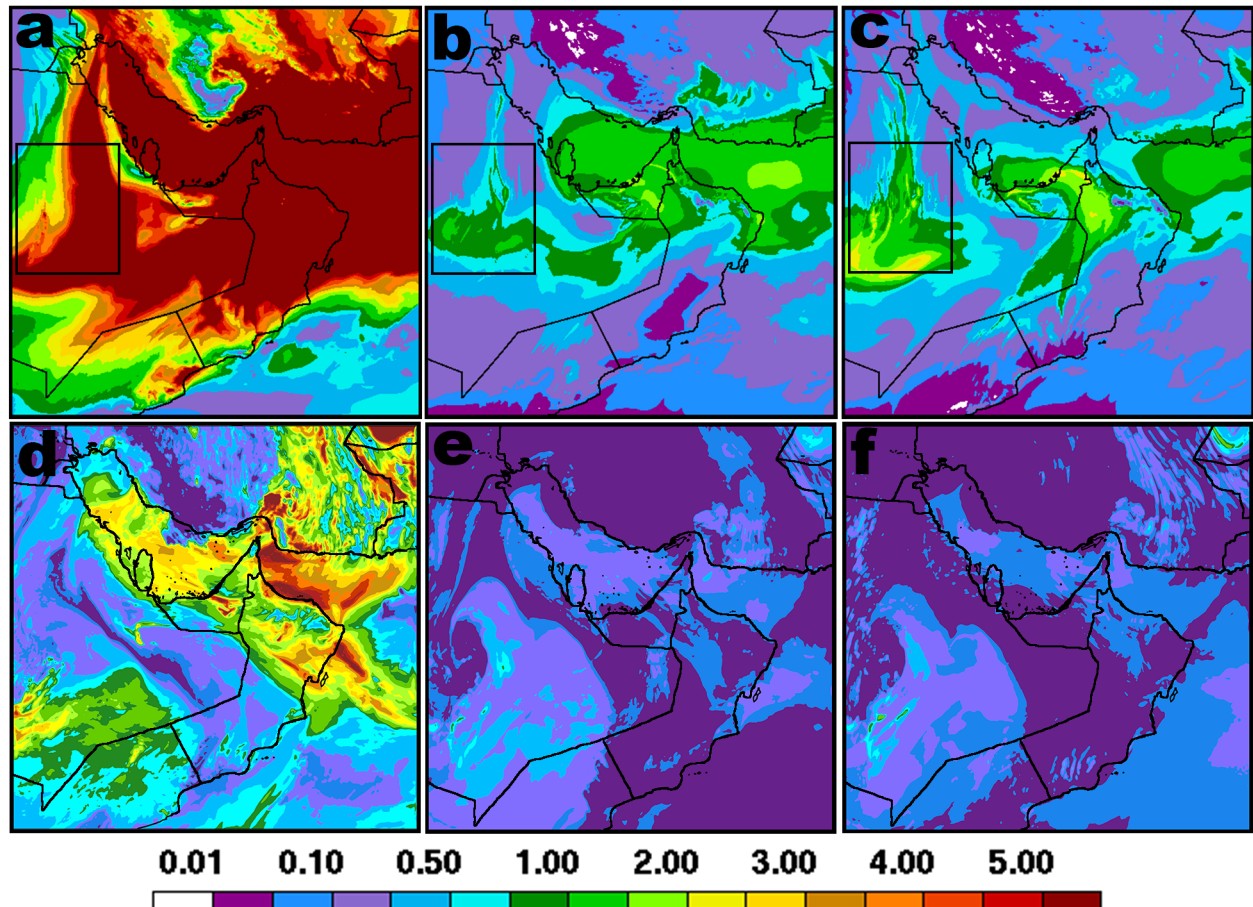

**Figure 6. Simulated total AOD at 550nm from (a-c) RAMS and (d-f) WRF-chem simulations at 0930 UTC 4 Aug, 2016 using the (a,d) Idealized dust lofting, (b,e) Ginoux dust sources, and (c,f) Walker dust sources. The black box over the central Arabian Peninsula denotes the analysis region shown in Figure 1.**

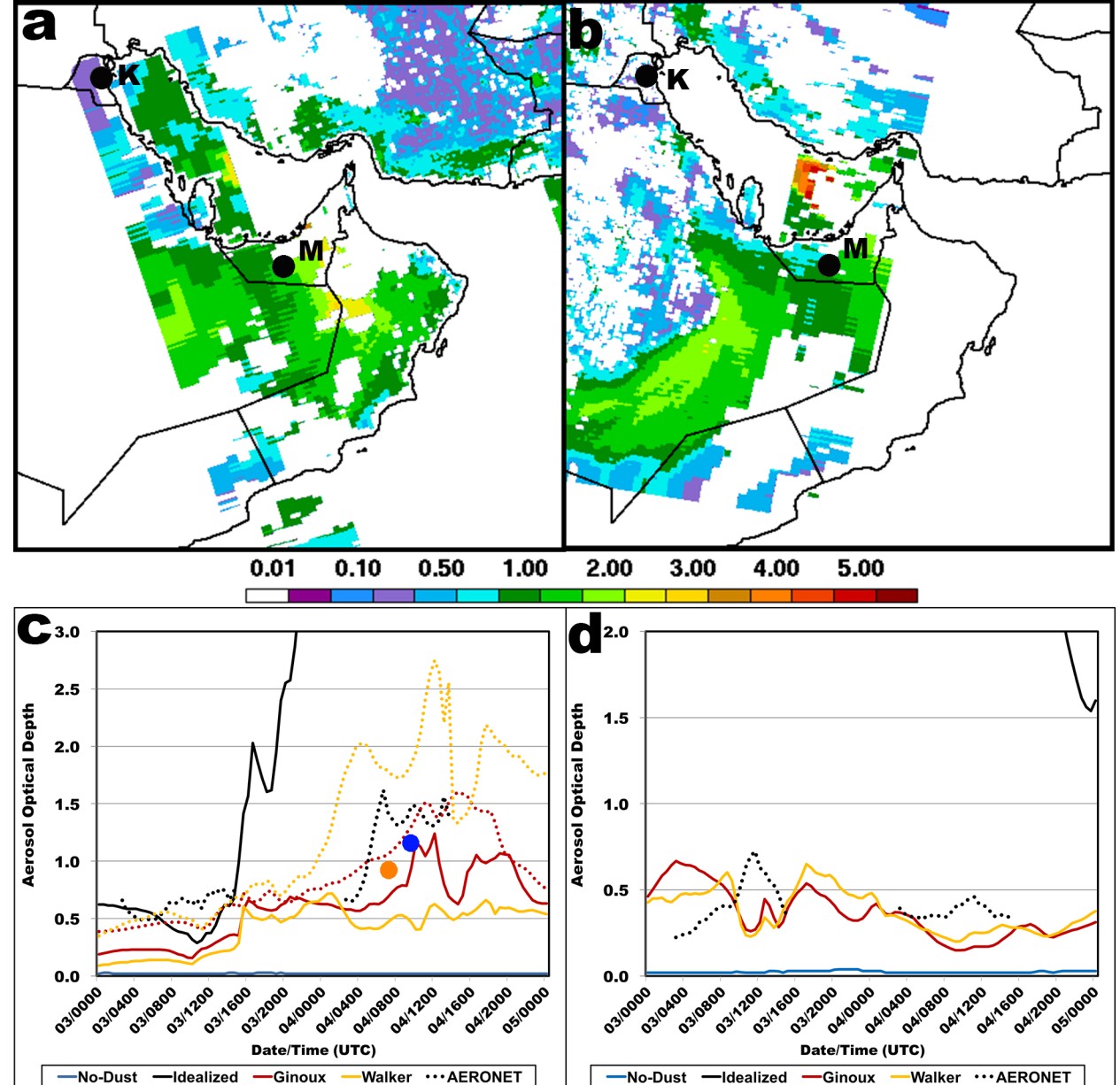

**Figure 7. (a) MODIS Aqua AOD at 550nm at 0915 UTC 4 Aug, 2016, (b) MODIS Terra AOD at 550nm at 0745 UTC 4 August 2016, (c,d) time series of AERONET AOD at 500nm and modeled time series of AOD at 550nm at the locations of (c) Mezaira, UAE (23.11N, 53.76E, level 1.5 data) and (d) Kuwait University, Kuwait (29.33N, 47.97E, level 1.0 data). The black dots in panels a and b indicate the locations of AERONET stations at Kuwait University (K) and Mezaira (M). Colored, dotted lines in panel c correspond to the time series for an in-plume location 2-degrees east of the Mezaira site as discussed in the text. The large blue (red) dot in panel c indicates the MODIS Aqua (Terra) AOD that has been spatially interpolated to the Mezaira location from the retrievals in panels a and b.**

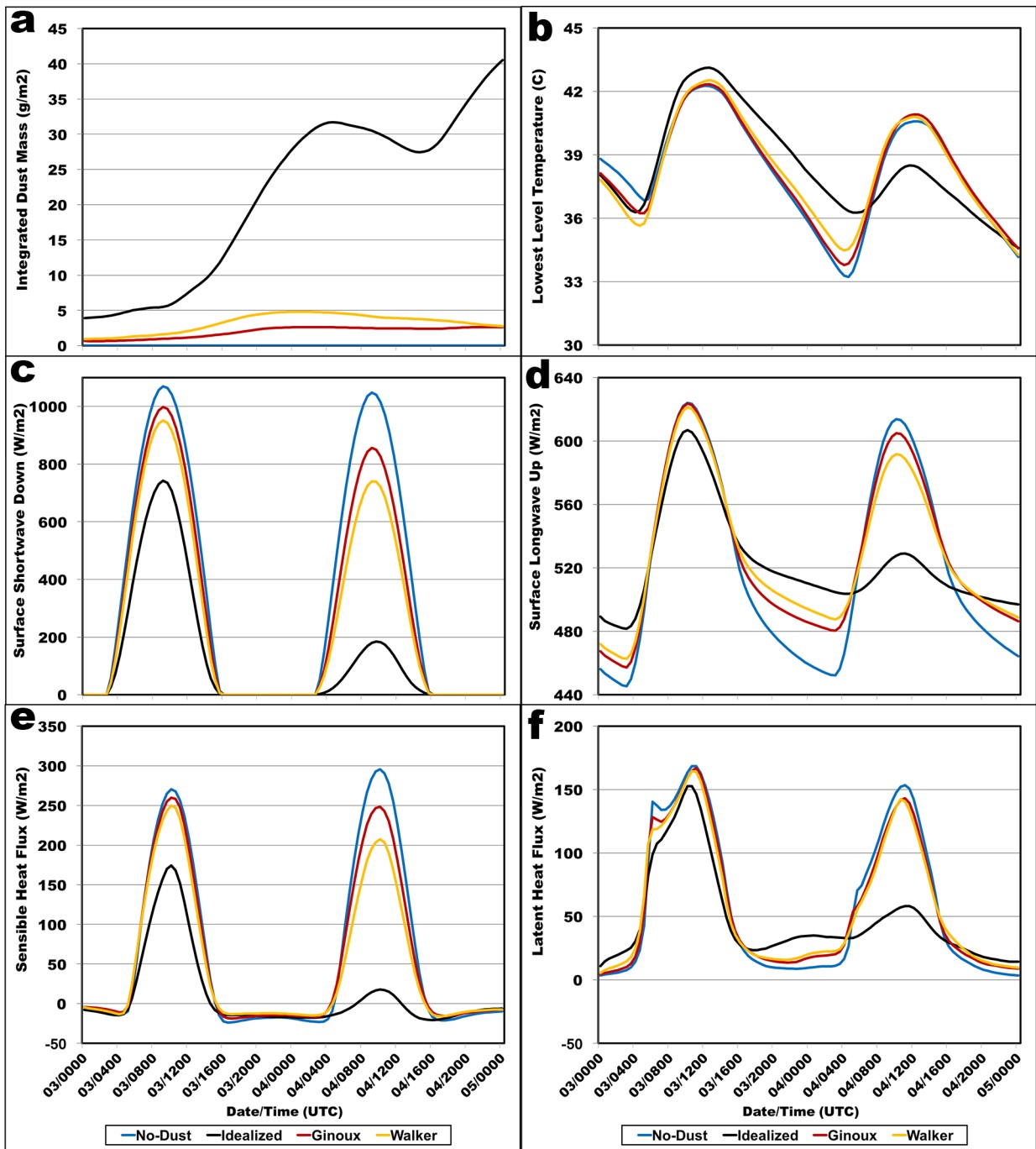

**Figure 8. Time series of (a) vertically integrated dust mass (g m⁻²), (b) lowest model level temperature (C), (c) surface downward shortwave radiation (W m⁻²), (d) surface upward longwave radiation (W m⁻²), (e) surface sensible heat flux (W m⁻²), and (f) surface latent heat flux (W m⁻²).**

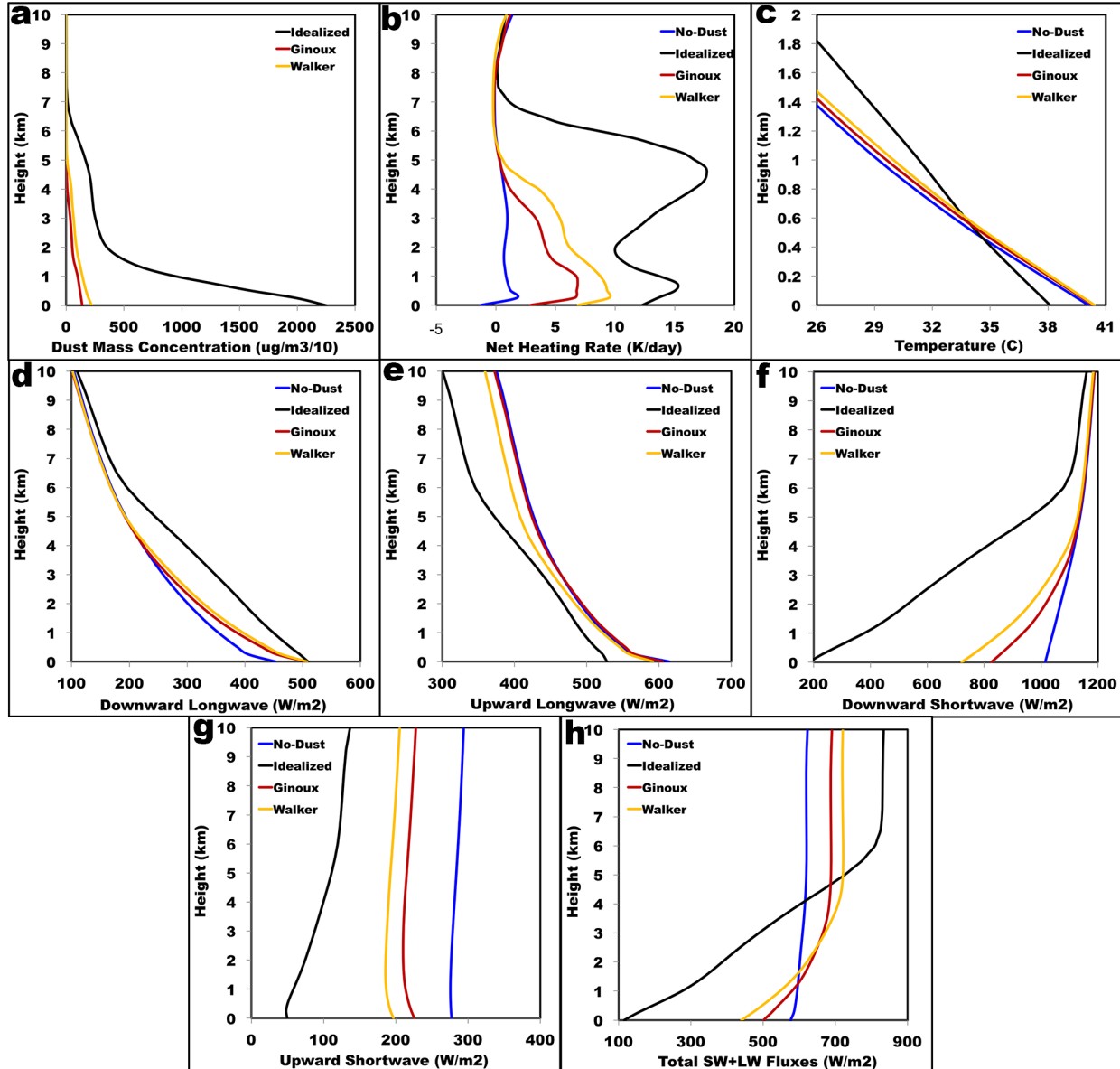

**Figure 9.** Daytime 1000 UTC (~1400 LST) 4 August vertical profiles of analysis-area averaged: (a) dust mass concentration (μg m$^{-3}$ 10$^{-1}$), (b) net radiative heating rate (K day$^{-1}$), (c) temperature (C), (d) downward longwave (W m$^{-2}$), (e) upward longwave (W m$^{-2}$), (f) downward shortwave (W m$^{-2}$), (g) upward shortwave (W m$^{-2}$), and (h) total radiative fluxes (W m$^{-2}$) computed as the sum of shortwave and longwave, downward minus upward fluxes. Note that panel (c) is on a different vertical scale so as to zoom in on lower-level temperature.

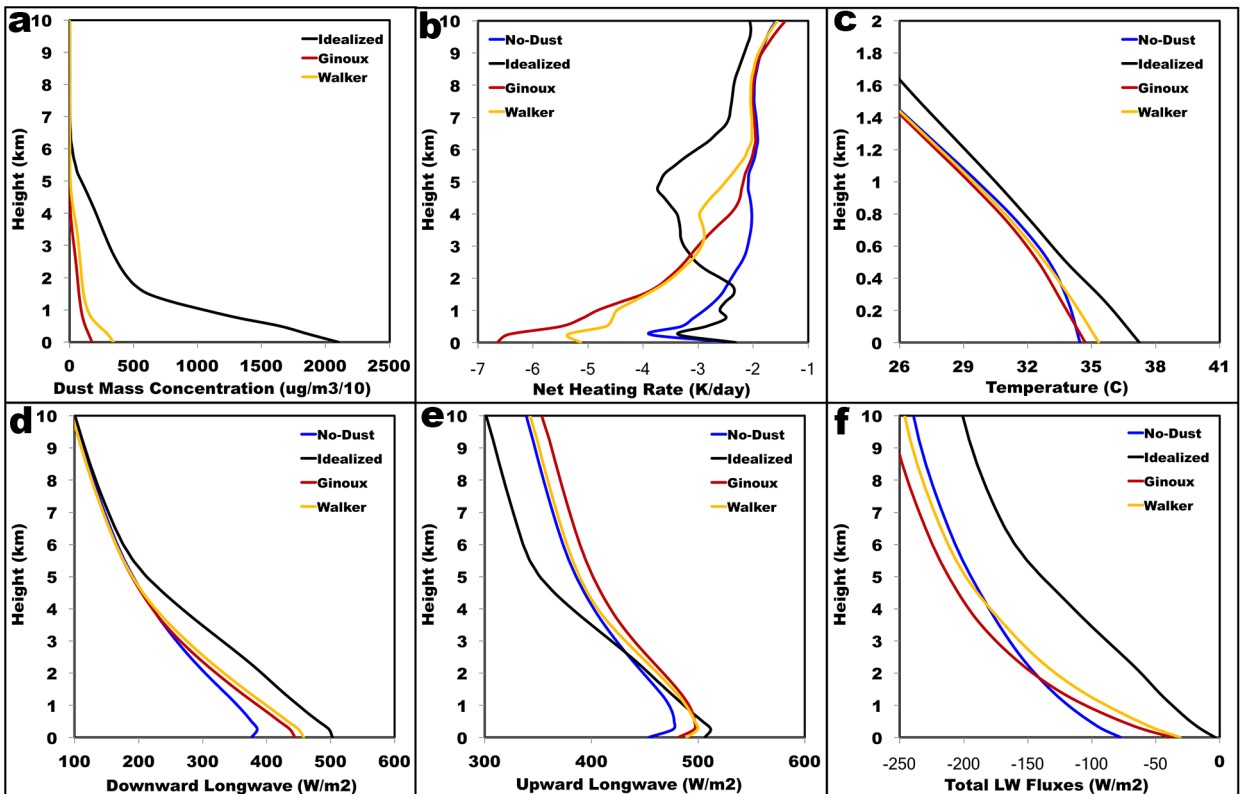

**Figure 10. Nighttime 0200 UTC (~0600 LST) 4 August vertical profiles of analysis-area-averaged: (a) dust mass concentration (µg m$^{-3}$ 10$^{-1}$), (b) net radiative heating rate (K day$^{-1}$), (c) temperature (C), (d) downward longwave (W m$^{-2}$), (e) upward longwave (W m$^{-2}$), and (f) total radiative fluxes (W m$^{-2}$) computed as the sum of shortwave and longwave, downward minus upward fluxes. Note that panel (c) is on a different vertical scale so as to zoom in on lower-level temperature.**