# Peer review of "The Influence of Simulated Surface Dust Lofting and Atmospheric Loading on Radiative Forcing"

_Atmospheric Chemistry and Physics, 2018_

## Referee Comment (RC1) · Anonymous Referee #1 · 5 Feb 2019

The authors present results of a regional NWP model simulations over the Arabian peninsula region, including mineral dust aerosol, for a case study during August 2015. They test sensitivity of dust simulations to two different models used, and to three different dust source representations. They compare these results to observations. One of the models is then used to examine the radiative effects of the dust in a cloud-free region, with a particular emphasis on radiative divergence, net radiative flux and vertical temperature profiles, contrasting the differences due to the different dust source representations.

The paper is mostly clearly and succinctly written, and easy to follow in terms of methodology and analysis. The interpretation of impacts of dust loading on radiative fluxes, vertical temperature profile, and surface fluxes are a valuable addition to the

literature and will help inform future studies on the potential impact of dust on cloud development. However, the earlier part of the paper (the impact of dust source representation on dust loading and AOD) is less clearly analysed and the main conclusions of this section are a little weak. The justification for including the 'idealized' lofting method is unclear (see major points below). The observations are not really sufficient to inform which of the two realistic lofting experiments (Ginoux and Walker) performs better and as such the first part of the paper is not particularly illuminating.

The abstract is fairly poor in describing the experiments the authors have conducted, why these were done, and their conclusions. A number of minor clarifications are necessary and are detailed below. However, with some additional clarifications and explanations the authors should be able to suitably address all these points and provide a paper suitable for ACP.

Major points

1) Idealized lofting

It is not clear why the authors choose to implement the 'idealized' lofting method when it generates such unrealistic results, and is also physically unrealistic. I suspect it is because this 'extreme' case becomes useful in section 4 when evaluating the radiative fluxes in terms of understanding how the system reacts to a 'kick.' Much more justification and explanation of the idealized method should be provided, as well as a statement that the authors do not expect it to respond realistically, and that it is retained for evaluation of 'extreme' purposes in section 4 (if that is the case). In terms of conclusions and abstract, it is not surprising that the idealized case produces inferior results – this is not a scientific finding.

2) Abstract

The abstract needs a complete re-write to follow a typical structure of description of a) the field/problem, b) description of experiments carried out and why, and c) results

found and their significance. Currently a) and b) are completely missing. Idealized lofting, if mentioned in the abstract, should be explained. It would be useful to relate 'extreme' and 'moderate' dust references to specific AOD ranges. L23-25 – this statement is not justified. The authors have not shown that the higher resolution source database produced better results (though the word 'detail' is ambiguous) – simply that it provided more spatial variability in the dust load. The fact that the Ginoux and Walker uplift experiments do not produce particularly different radiative effects should be stated (and also discussed in the paper).

3) Significance of Section 3

Overall the observational evidence for evaluating the Ginoux vs Walker uplift experiments is fairly weak. The Walker simulation provides much greater spatial variability due to the higher resolution of the input surface data compared to the Ginoux dataset. However, the sparsity of data over the region prevents the authors from reliably evaluating whether one dataset is better than the other. The MODIS data shown is rather patchy and also only shown for part of the simulation region. The AERONET data is not conclusive in the evaluation and a small offset in model analysis region for the AERONET comparison produces significantly different results. The authors should either attempt to expand their observational comparison to inform the model comparison, or if this is not possible, modify the text and conclusions appropriately to say that lack of observations prevented a proper evaluation of the two dust source datasets. Even without being able to say which dataset is better, it is a useful finding that more resolution in surface dust source areas translates to more spatial variability in the atmosphere, even after several (?) days of transport.

4) Comparison against literature

There is rather little comparison against other literature in general – this would add to the significance of the article – both in the context of implementing different dust source maps, and in terms of the radiative effect (Section 4) results.

Minor points and clarifications

Title – I encourage the authors to make this clearer – e.g. remove 'erodible fraction' and possibly include 'and atmospheric loading' before 'radiative forcing'

P2L9 – dust can cause atmospheric cooling in the LW also

Section 2.2 – GOCART should be briefly described (e.g. size bins, uplift scheme) to give the equivalent information provided on the dust scheme in RAMS.

Section 2.3, p5L27 onwards – Does this mean that the erodible fraction over the whole land-domain is 100%? Please clarify.

Section 2.3 p5 L27-35 – more background should be provided on each of the 3 surface lofting methodologies/datasets, since this is a key process and result within the paper. E.g. How were the datasets produced? What are they based on? Why are they different? Is the Ginoux dataset the topographic low source function?

Figure 2a – why are there lines around some of the grid boxes? Is this an artefact? It seems unphysical.

P7L3 – refs to fig 4a – it's pretty difficult to see the dust over the desert. It would be helpful to refer the reader to the AOD figure 7 here too (see also comment about domain shown in fig 7).

P7L8-9 – could this also be the higher resolution between the reanalysis and the model runs?

P7L11-19 – The inclusion of the NAAPS plot is confusing and unhelpful. The inclusion of data from NAAPS is sudden and unexplained. Comparing a model to another model is not helpful. I suggest removing the NAAPS figure and text completely. It does not add anything to the paper.

P7 L31-32 – 'In both models, the Walker simulation captures more dust mass detail with respect to the lofting locations due to the precise, high resolution nature of the

database.' – This should be reworded. The simulation may show more 'detail' – (spatial variability is probably a better word) but there are no constraints to show that this is correct. Due to the source database being higher resolution, one would expect the atmospheric dust loading to be more spatially variable. This does not show it is better or correct though.

P7 l30 onwards – WRF results are quite different to RAMS – the authors should discuss this and attempt to explain why.

P8 L1-2 – See above points about NAAPS – no need for NAAPS data here. Actual observations should be used to verify simulations, not another model! (And if there are no observations, a simple statement to this effect is sufficient).

P8 L5 – does this mean that RAMS does not include radiative feedbacks of dust, onto dynamics, etc.?

P8 L8 – refractive index at which wavelength? Assuming this is 500-550nm, the imaginary part is relatively high (e.g. see Song et al., 2018, Balkanski et al., 2017). This will impact the radiative results in section 4 by causing increased absorption and atmospheric heating, and should be discussed. E.g. Strong et al. (2018) show that small changes in optical properties can have huge effects on circulation.

P8L9 'spheroid-like index of refraction' – clarify this – index of refraction does not have a shape.

P8 L8-10 – what refractive index in the LW is used?

P8 L15-16 – what dust optical properties are used in WRF?

P9 L1 – 'similar predicted synoptic situations' – this doesn't seem justified – the streamlines are quite different between RAMS and WRF – and dust uplift is extremely sensitive to small differences in wind pattern, speed and strength.

P9 L3 – 'trends' – which ones? The authors have only discussed differences between

idealized vs Ginoux/Walker, not Ginoux vs Walker, which are clearly not the same for WRF and RAMS.

P9 L24-27 – and also impacts the Walker expt more because there is more spatial variability in the atmospheric dust load?

P12 L33 – 'small warming' – how much?

Section 4 – there is no comparison between Ginoux vs Walker results here – why not?

Section 4 – are the radiative results consistent with other work? E.g. Marsham et al. (2016)?

Conclusion – more text should be added to cover the results of the source dataset experiments – e.g. the effects of Walker vs Ginoux simulations, and the fact that the Walker simulations produced more patchy dust loadings than Ginoux.

Figures – take care that the same country boundaries are shown on all maps. E.g. fig 3 – the WRF plots show different country boundaries to the other plots. H and I do not show boundaries. Check ACP guidelines for international borders.

Figures 5-6 – the authors should show the analysis region on figures a-c

Fig 7 – why is the same geographical domain as figs 5-6 not shown? A larger area would be more appropriate, especially since the radiative analysis region is not even covered in fig 7.

References

Balkanski, Y., et al.: Reevaluation of Mineral aerosol radiative forcings suggests a better agreement with satellite and AERONET data, Atmos. Chem. Phys., 7, 81-95, https://doi.org/10.5194/acp-7-81-2007, 2007.

Marsham, J. H., et al.: The contrasting roles of water and dust in controlling daily variations in radiative heating of the summertime Saharan heat low, Atmos. Chem.

Phys., 16, 3563-3575, https://doi.org/10.5194/acp-16-3563-2016, 2016.

Song, Q., et al.: Net radiative effects of dust in the tropical North Atlantic based on integrated satellite observations and in situ measurements, Atmos. Chem. Phys., 18, 11303-11322, https://doi.org/10.5194/acp-18-11303-2018, 2018.

Strong, J. D. O., Vecchi, G. A., Ginoux, P. (2018). The climatological effect of Saharan dust on global tropical cyclones in a fully coupled GCM. Journal of Geophysical Research: Atmospheres, 123. https://doi.org/10.1029/2017JD027808

---

## Referee Comment (RC2) · Anonymous Referee #2 · 11 Feb 2019

The manuscript analyses the numerical simulations of dust lofting using erodible dust fraction as input and its impact on radiation during daytime hours and nighttime hours. The dust erodible fraction is taken from dataset from three methods, namely, the "Idealized", "Ginoux", and "Walker". The numerical simulations are done with WRF and RAMS over the Arabian peninsula. Overall, the manuscript is well written, logically presented, and is interesting to read. I recommend the publication of this manuscript after considering the following suggestions:

1. I could not find any quantitative validation exercise between MODIS and Model AOD. Please clarify. Can the MODIS AOD be extracted at some of the stations and compared with Model data? It has also been inferred in previous studies that MODIS data overpredicts AOD for regions predominant with dust (see Remer et al., 2005).
Please take this into account while validation of the model. Remer LA, Kaufman YJ, Tanré D, Matto S, Chu DA, Martins JV, Li RR, Ichoku C, Levy RC, Kleidman RG, Eck TF, Vermote E, Holben BN (2005) The MODIS aerosol algorithm, products, and validation. J Atmos Sci 62:947–973. https://doi.org/10.1175/JAS3385.1 2. A large underestimation is seen between model and AERONET AOD. What could be the reason for this? It will be nice if the authors could provide a quantitative validation, including bias and normalised mean error. How much is the uncertainty in AERONET AOD for regions predominant with dust? I suggest strengthening this Section by providing information from any available literature study as well. One of such studies, I recently found is by Kokkalis et al., (2018). Long-Term Ground-Based Measurements of Aerosol Optical Depth over Kuwait City. Remote Sensing, 10, 1807; DOI:10.3390/rs1011180710. 3. Also, why "Ginoux"' is larger than the "Walker" (refer to Figure 7c)? Please include some discussion on this. 4. How much is the difference between the simulated dust concentration from NAAPS and that from RAMS and WRF? I suggest the authors discuss this as they provide NAAPS dust concentration. 5. How much is the expected uncertainty in your model values for radiative impacts? 6. I suggest to compare the radiative implications, such as radiative cooling/heating during daytime and nighttime with observational data 7. Refer to Figure 10f: Is this for Total LW fluxes? Or for total radiative fluxes (SW+LW)? Please check.
* * *

---

## Author Comment (AC1) · 29 May 2019

**Reviewer comments below are in standard black font, while the author responses are in blue italic font for contrast.**

General reply to reviewers based on overarching comments:

*We thank the reviewers for their time in examining our manuscript and offering constructive criticism, comments, and suggestions. We feel that reviewer comments have led to an improved manuscript. As will be discussed in detail below in response to specific comments and questions, this paper presents a theoretical modeling study placed in the context of a dust lofting event over the Arabian Peninsula that explores the potential radiative response to variable dust loading using dust lofting models and dust-sensitive radiation schemes embedded within sophisticated high-resolution model environments. The main goal of the paper is to examine the mean differences in radiative quantities and atmospheric temperature resulting from differences in dust loading that result from applying different dust erodible fraction datasets to the lofting model.*

*While the Arabian Peninsula is well-known for its expansive dust storms, few dust lofting studies have been performed over this region. This is, perhaps, because aerosol related data in this region are limited. As such, we have provided a more qualitative model comparison to the limited aerosol observations in the area in order to broadly demonstrate that one of the models (RAMS) does a favorable job in simulating dust lofting when the dust erodible fraction is constrained by geographical datasets, while noting that precisely simulating the magnitude and location of individual dust plumes is incredibly difficult. Following this, the RAMS model was then used to investigate dust radiative effects in the simulated environment. It is not our intent to determine which dataset leads to the best model representation of dust lofting. Walker et al. (2009) provide such an assessment with regards to dust lofting and surface visibility. Our focus is on determining the potential range of dust radiative effects by comparing a simulation with no-dust to those with varying amounts of dust generated by use of different specifications of surface dust erodible fraction.*

*Overall, we have worked to more clearly frame the focus of this paper as a theoretical examination of dust radiative effects in a case study context, while noting that dust AOD observations are limited, yet they compare favorably to RAMS simulations when dust erodible fraction appropriately constrains the amount of lofting.*

**Anonymous Referee #1**

The authors present results of a regional NWP model simulations over the Arabian Peninsula region, including mineral dust aerosol, for a case study during August 2015. They test sensitivity of dust simulations to two different models used, and to three different dust source representations. They compare these results to observations. One of the models is then used to

examine the radiative effects of the dust in a cloud-free region, with a particular emphasis on radiative divergence, net radiative flux, and vertical temperature profiles, contrasting the differences due to the different dust source representations.

The paper is mostly clearly and succinctly written, and easy to follow in terms of methodology and analysis. The interpretation of impacts of dust loading on radiative fluxes, vertical temperature profile, and surface fluxes are a valuable addition to the literature and will help inform future studies on the potential impact of dust on cloud development. However, the earlier part of the paper (the impact of dust source representation on dust loading and AOD) is less clearly analyzed and the main conclusions of this section are a little weak. The justification for including the "idealized" lofting method is unclear (see major points below). The observations are not really sufficient to inform which of the two realistic lofting experiments (Ginoux and Walker) performs better and as such the first part of the paper is not particularly illuminating.

*We thank this reviewer for your overall assessment of this manuscript. Our general reply to reviewers at the top of this document is meant to provide focus on the intent of the paper while addressing the concerns regarding the comparisons to limited observations. We have revised the manuscript to help focus the direction and intent of the paper and address the utility of the Idealized lofting experiment, as noted above.*

The abstract is fairly poor in describing the experiments the authors have conducted, why these were done, and their conclusions. A number of minor clarifications are necessary and are detailed below. However, with some additional clarifications and explanations the authors should be able to suitably address all these points and provide a paper suitable for ACP.

*We have examined the abstract and have rewritten it to better describe the motivation, experiments, and conclusions. We have also responded to each specific comment below.*

Major points

1) Idealized lofting
It is not clear why the authors choose to implement the 'idealized' lofting method when it generates such unrealistic results, and is also physically unrealistic. I suspect it is because this 'extreme' case becomes useful in section 4 when evaluating the radiative fluxes in terms of understanding how the system reacts to a 'kick'. Much more justification and explanation of the idealized method should be provided, as well as a statement that the authors do not expect it to respond realistically, and that it is retained for evaluation of 'extreme' purposes in section 4 (if that is the case). In terms of conclusions and abstract, it is not surprising that the idealized case produces inferior results – this is not a scientific finding.

*We noted in the discussion of the simulations that the "Idealized" lofting method was included as one of our experiments since this method has been used in another study that simulates idealized conditions (e.g. Seigel and van den Heever 2012). In idealized simulations the "Idealized" lofting method, that can loft dust in any grid cell containing dry soil, certain clay fractions, and low vegetation, has been shown to produce reasonable amounts of dust for localized dust events. It seems fair to extend this to a case study for testing to examine the upper*

*end of potential dust lofting, even if this may be unrealistic. As such, we have revised the manuscript to present this as being an upper limit to dust lofting that could occur in this model setting, and then examine the upper limit of radiative response. We have modified the text to better clarify this motivation.*

2) Abstract

The abstract needs a complete re-write to follow a typical structure of description of a) the field/problem, b) description of experiments carried out and why, and c) results found and their significance. Currently a) and b) are completely missing. Idealized lofting, if mentioned in the abstract, should be explained. It would be useful to relate 'extreme' and 'moderate' dust references to specific AOD ranges. L23-25 – this statement is not justified. The authors have not shown that the higher resolution source database produced better results (though the word 'detail' is ambiguous) – simply that it provided more spatial variability in the dust load. The fact that the Ginoux and Walker uplift experiments do not produce particularly difference radiative effects should be stated (and also discussed in the paper).

*The abstract has been re-written to provide a concise summary of the work presented in this manuscript. Also, we have added discussion regarding the similarities in the results comparing the Ginoux and Walker experiments.*

3) Significance of Section 3

Overall the observational evidence for evaluating the Ginoux vs. Walker uplift experiments is fairly weak. The Walker simulation provides much greater spatial variability due to the higher resolution of the input surface data compared to the Ginoux dataset. However, the sparsity of the data over the region prevents the authors from reliably evaluating whether one dataset is better than the other. The MODIS data shown is rather patchy and also only show for part of the simulation region. The AERONET data is not conclusive in the evaluation and a small offset in model analysis region for the AERONET comparison produces significantly different results. The authors should either attempt to expand their observational comparison to inform the model comparison, or if this is not possible, modify the text and conclusions appropriately to say that lack of observations prevent a proper evaluation of the two dust source datasets. Even without being able to say which dataset is better, it is a useful finding that more resolution in surface dust source area translates to more spatial variability in the atmosphere, even after several days of transport.

*Thank you for this comment. We agree that it would be desirable to have a more extensive AERONET array and better MODIS coverage. However, we have presented what limited observations are available for comparing dust. We have modified the text to highlight that observations are limited and thus our observational comparison is intended to be qualitative in nature. The single southern AERONET site provides us with only a single point comparison near the UAE / Persian Gulf dust plume. Performing grid point comparisons between models and observations often provides limited utility in events where key features, such as dust plumes, are slightly displaced in the simulations. In our case the simulated dust plume over the UAE and Persian Gulf is slightly displaced, but magnitudes of AOD are similar to the in-plume MODIS AODs. We have added text that addresses the limited nature of the observations and their comparisons to model results.*

4) Comparison against literature
There is rather little comparison against other literature in general – this would add to the
significance of the article – both in the context of implementing different dust source maps, and
in terms of the radiative effect (Section 4) results.

*Throughout the manuscript we have added more comparison between the results of this study
and past work including some comparisons with the following papers: Slingo et al. (2006), Shell
et al. (2007), Lau and Kim (2007), Marsham et al. (2016), Hansell et al. (2010), Kosmopoulos et
al. (2017).*

Minor points and clarifications

Title – I encourage the authors to make this clearer – e.g. remove 'erodible fraction' and possibly
include 'and atmospheric loading' before 'radiative forcing'

*We have changed the title to remove "erodible fraction" and include "and atmospheric
loading".*

P2L9 – dust can cause atmospheric cooling in the LW also

*We have added a statement here to the effect that LW emission in the dust layer adds a cooling
tendency within the dust layer, but warming effect via LW emission adjacent to the dust layer
(e.g. Slingo et al. 2006; Wang et al. 2013).*

Section 2.2 – GOCART should be briefly described (e.g. size bins, uplift scheme) to give the
equivalent information provided on the dust scheme in RAMS.

*We have added some text to section 2.1 to indicate that RAMS' dust scheme is largely based on
GOCART with some additional modifications related to soil type, vegetation, and dust lofting
size bins. WRF-Chem uses GOCART dust lofting. The details of GOCART dust lofting can be
found in Ginoux et al. (2001) as referenced.*

Section 2.3 – p5 L27 onwards – Does this mean that the erodible fraction over the whole land-
domain is 100%? Please clarify.

*In the Idealized simulation, the erodible fraction over the whole land domain is 1.0 (100%). This
was done for the Seigel and van den Heever (2012) limited area domain and produced quite
favorable dust amounts over a limited time frame involving outflows from deep convection. We
have added text to section 2.3 to clarify this. We have also placed the Idealized simulation in the
context of representing the expected upper bound on dust lofting in this type of case study. We
found it quite informative to know the potential upper limit of radiative effects that could be
expected within the given modeling framework and parameterization.*

Section 2.3 – p5 L27-35 – more background should be provided on each of the 3 surface lofting
methodologies/datasets, since this is a key process and result within the paper. E.g. How were

the datasets produced? What are they based on? Why are they different. Is the Ginoux dataset the topographic low source function?

*We have added into this section several sentences that clarify the application of the Idealized lofting method, the Ginoux method based on topographic depressions, and the Walker method based on manual satellite identification of dust lofted areas. Each of these methods has an associated citation for which the referenced paper can provide the intricate details of the lofting methods/databases.*

Figure 2a – why are there lines around some of the grid boxes? Is this an artefact? It seems unphysical.

*The lines are just an artifact of the plotting tool and the discrete application of Ginoux 1-deg gridded dust sources to the model grid.*

P7L3 – refs to Fig 4a – it's pretty difficult to see the dust over the desert. It would be helpful to refer the reader to AOD figure 7 here too (see also comment about domain shown in fig 7).

*The dust over the desert is, indeed, difficult to discern in the visible imagery due to the similar colors of the dust and land area. We have updated the text to also point to the MODIS imagery in Figure 7 that shows some of the dust presence associated with the two plumes.*

P7L8-9 – could this also be the higher resolution between the reanalysis and the model runs?

*The differences in the magnitude of the 1000mb temperature field between the reanalysis and model data are probably more the result of the differences in the representation of topography and the land surface parameterizations between the models used here and the model portion of the reanalysis technique. The differences in the horizontal variability and spatial details between the models and reanalysis are likely due to resolution differences. We have added text in the manuscript to clarify these differences.*

P7L11-19 – The inclusion of the NAAPS plot is confusing and unhelpful. The inclusion of data from NAAPS is sudden and unexplained. Comparing a model to another model is not helpful. I suggest removing the NAAPS figure and text completely. It does not add anything to the paper.

*We have removed all discussion and figures related to NAAPS.*

P7L31-32 – 'In both models, the Walker simulations captures more dust mass detail with respect to the lofting locations due to the precise, high resolution nature of the database.' – This should be reworded. The simulation may show more 'detail' – (spatial variability is probably a better word) but there are no constraints to show that this is correct. Due to the source database being higher resolution, one would expect the atmospheric dust loading to be more spatially variable. The does not show it is better or correct though.

*We have restated this sentence to note that the high-resolution Walker dust source database leads to the generation of comparatively greater fine-scale spatial variability in lofted dust in*

*association with known dust source locations. We have also added a statement that while there is increased precision in lofted locations with the Walker database, that does not imply that the net amount of lofted dust is more accurate than that lofted via the Ginoux database. Walker et al. (2009) provide such an assessment.*

P7L30 onwards – WRF results are quite different to RAMS – the authors should discuss this and attempt to explain why.

*Yes, WRF and RAMS dust amounts and AOD are quite different. We state at the end of Section 3.1 that these differences exist and that there is a separate study under way to perform an extensive model inter-comparison involving RAMS, WRF, and another model as well. This type of model inter-comparison is beyond the scope of this paper and will appear in a separate manuscript in the future. However, we have also added a statement to the end of Section 3.1 which says that both RAMS and WRF use the dust lofting techniques of the GOCART model (Ginoux et al. 2001) and the same erodible fraction databases; as such, we speculate that the prediction of the near-surface wind speed, the soil moisture, dust deposition rates, and dust binning may all be playing a role in contributing to the differences. A separate in-depth study will help shed light on this.*

P8L1-2 – See above points about NAAPS – no need for NAAPS data here. Actual observations should be used to verify simulations, no another model! (And if there are no observations, a simple statement to this effect is sufficient).

*We have removed NAAPS from the paper and have noted in the paper the limited aerosol observations available for comparison.*

P8L5 – does this mean that RAMS does not include radiative feedbacks of dust, onto dynamics, etc.?

*No. It means that we used an offline model to compute diagnostic AOD for comparison with MODIS and AERONET. The RAMS model does not provide AOD as a standard output diagnostic, thus we had to generate this offline. However, the aerosols are radiatively interactive, thus, impacting the radiation flux profiles and providing feedbacks to the dynamics and thermodynamics. We have added a statement and reference in this regards in the section that describes the RAMS aerosol model.*

P8L8 – refractive index at which wavelength? Assuming this is 500-550nm, the imaginary part is relatively high (e.g. see Song et al. 2018, Balkanski et al. 2017). This will impact the radiative results in section 4 by causing increased absorption and atmospheric heating, and should be discussed. E.g. Strong et al. (2018) show that small changes in optical properties can have huge effects on circulation.

*This index of refraction is referring to the 550nm mentioned above on line 5 for the offline analysis of AOD. Per this reviewer question, we have added additional text and citations in section 2.1 regarding the use of a dust complex index of refraction of 1.53+0.0015i for dust for*

*wavelengths up to ~2000nm wavelength for generating RAMS lookup tables of aerosol optical properties. Further, we note that AOD is not sensitive to the imaginary index of refraction.*

P8L9 – 'spheroid-like index of refraction' – clarify this – index of refraction does not have a shape.

*The wording has been changed in the text to clarify the assignment of the dust index of refraction used for computing AOD from our offline model of aerosol extinction.*

P8L8-10 – what refractive index in the LW is used?

*The AOD analysis was only done at 550nm. However, we have added text to Section 2.1 to better describe the assignment of the indices of refraction for dust at various wavelengths. As noted in a response above we use in RAMS a complex index of refraction of 1.53+0.0015i for dust up to ~2000nm. We state that Stokowski (2005) provides a plot of refractive index as it varies with wavelength in the RAMS model.*

P8L15-16 – what dust optical properties are used in WRF?

*We have added a statement in the text indicating that the dust real index of refraction for computing AOD at 550nm in WRF is set at 1.53.*

P9L1 – 'similar predicted synoptic situations' – this doesn't seem justified – the streamlines are quite different between RAMS and WRF – and dust uplift is extremely sensitive to small differences in wind pattern, speed and strength.

*We have modified this section and section 2.4 to better state the similarities and differences between the synoptic fields shown in figure 3. The streamlines are shown so as to demonstrate that both models produced the northerly flow associated with the Saudi dust plume and the southerly to south-westerly flow associated with the UAE plume. We have added statements in the text that address the differences in AOD between RAMS and WRF and offer speculation that differences in wind speed and other conditions could explain the differences in dust lofting between the models. As noted earlier, this involves an on-going model inter-comparison study for a separate manuscript.*

P9L3 – 'trends' – which ones? The authors have only discussed differences between idealized vs. Ginoux/Walker, not Ginoux vs. Walker, which are clearly not the same for WRF and RAMS.

*We have reworded this paragraph to better summarize the overarching differences between simulations and models and between modeled and observed dust AOD.*

P9L24-27 – and also impacts the Walker expt more because there is more spatial variability in the atmospheric dust load?

*We have added discussion throughout the manuscript regarding the differences between simulated Ginoux and Walker dust plume concentration and AOD. We specifically discuss that*

*the widespread, small erodible fraction with Ginoux dust data tend to produce more broad dust plumes with lower maximum AOD. The Walker data tend to generate more focused plumes with higher maximum AOD and greater spatial variability. This certainly impacts the interpretation of the grid point comparisons to AERONET sites.*

P12L33 – 'small warming' – how much?

*We have added the detail that the small warming is ~0.3-0.4C for the Ginoux and Walker simulations compared to No-Dust.*

Section 4 – there is no comparison between Ginoux vs. Walker results here – why not?

*While some comparisons are made between the Ginoux and Walker results, they are noted as being quite similar compared to the Idealized simulation. Many places throughout section 4 indicate monotonic changes in radiative fluxes with dust loading, and we have noted that the mean dust loading in the analysis region increases from the Ginoux to Walker to Idealized simulations (see figure 9a dust profiles). The discussion of monotonic changes implicitly compares all three dust-lofting simulations to the No-Dust simulation and to each other. However, in the revised manuscript we have included additional statements to compare the Ginoux and Walker simulations.*

Section 4 – are the radiative results consistent with other work? E.g. Marcham et al. (2016)?

*We noted in section 4.2 that the reductions in shortwave radiation for the Ginoux and Walker simulations are similar to those seen in Slingo et al. (2006) and Kosmopoulos et al. (2017) for similar AOD. We added more detail to this to note that both studies show surface shortwave reductions of approximately 200-250W/m2 for dust AOD on the order of 1.5-2.5.*

*We have added more comparisons to past work that is comparable to this study including comparisons to shortwave and longwave fluxes as well as estimated heating rates. (e.g. Marsham et al. 2016, Hansell et al. 2010).*

Conclusion – more text should be added to cover the results of the source dataset experiments – e.g. the effects of Walker vs. Ginoux simulations, and the fact that the Walker simulations produced more patchy dust loadings than Ginoux.

*We have updated the conclusions to offer a more comprehensive summary of the results and include more summary of the differences between the Ginoux and Walker simulations.*

Figures – take care that the same country boundaries are shown on all maps. E.g. fig 3 – the WRF plots show different country boundaries to the other plots. H and I do not show boundaries. Check ACP guidelines for international borders.

*We have worked to make the country boundaries similar among the RAMS and WRF plots.*

Figures 5-6 – the authors should show the analysis region on figures a-c

*We have added the analysis region box to figures 5&6 panels a-c.*

Fig 7 – why is the same geographical domain as figs 5-6 not shown? A larger area would be more appropriate, especially since the radiative analysis region is not even covered in fig 7.

*The model AOD (figure 6) is available over the full simulation domain, but the MODIS AOD (figure 7) does not cover the full model domain, but rather a limited swath. We had zoomed in over the plumes to be able to see some of the higher AOD pixels associated with the Persian Gulf plume. However, we have modified the figure to show the full domain, which does provide a better view of the Saudi dust plume.*

References

Balkanski, Y., et al.: Reevaluation of Mineral aerosol radiative forcings suggests a better agreement with satellite and AERONET data, Atmos. Chem. Phys., 7, 81-95, https://doi.org/10.5194/acp-7-81-2007, 2007.

Marsham, J. H., et al.: The contrasting roles of water and dust in controlling daily variations in radiative heating of the summertime Saharan heat low, Atmos. Chem. Phys., 16, 3563-3575, https://doi.org/10.5194/acp-16-3563-2016, 2016.

Song, Q., et al.: Net radiative effects of dust in the tropical North Atlantic based on integrated satellite observations and in situ measurements, Atmos. Chem. Phys., 18, 11303-11322, https://doi.org/10.5194/acp-18-11303-2018, 2018.

Strong, J. D. O., Vecchi, G. A., Ginoux, P. (2018). The climatological effect of Saha- ran dust on global tropical cyclones in a fully coupled GCM. Journal of Geophysical Research: Atmospheres, 123. https://doi.org/10.1029/2017JD027808

---

## Author Comment (AC2) · 29 May 2019

**Reviewer comments below are in standard black font, while the author responses are in blue italic font for contrast.**

General reply to reviewers based on overarching comments:

*We thank the reviewers for their time in examining our manuscript and offering constructive criticism, comments, and suggestions. We feel that reviewer comments have led to an improved manuscript. As will be discussed in detail below in response to specific comments and questions, this paper presents a theoretical modeling study placed in the context of a dust lofting event over the Arabian Peninsula that explores the potential radiative response to variable dust loading using dust lofting models and dust-sensitive radiation schemes embedded within sophisticated high-resolution model environments. The main goal of the paper is to examine the mean differences in radiative quantities and atmospheric temperature resulting from differences in dust loading that result from applying different dust erodible fraction datasets to the lofting model.*

*While the Arabian Peninsula is well-known for its expansive dust storms, few dust lofting studies have been performed over this region. This is, perhaps, because aerosol related data in this region are limited. As such, we have provided a more qualitative model comparison to the limited aerosol observations in the area in order to broadly demonstrate that one of the models (RAMS) does a favorable job in simulating dust lofting when the dust erodible fraction is constrained by geographical datasets, while noting that precisely simulating the magnitude and location of individual dust plumes is incredibly difficult. Following this, the RAMS model was then used to investigate dust radiative effects in the simulated environment. It is not our intent to determine which dataset leads to the best model representation of dust lofting. Walker et al. (2009) provide such an assessment with regards to dust lofting and surface visibility. Our focus is on determining the potential range of dust radiative effects by comparing a simulation with no-dust to those with varying amounts of dust generated by use of different specifications of surface dust erodible fraction.*

*Overall, we have worked to more clearly frame the focus of this paper as a theoretical examination of dust radiative effects in a case study context, while noting that dust AOD observations are limited, yet they compare favorably to RAMS simulations when dust erodible fraction appropriately constrains the amount of lofting.*

**Anonymous Referee #2**

The manuscript analyses the numerical simulations of dust lofting using erodible dust fraction as input and its impact on radiation during daytime hours and nighttime hours. The dust erodible faction is taken from dataset from three methods, namely, the "idealized", "Ginoux", and "Walker". The numerical simulations are done with WRF and RAMS over the Arabian

Peninsula. Overall, the manuscript is well written, logically presented, and is interesting to read. I recommend the publication of this manuscript after considering the following suggestions:

1. I could not find any quantitative validation exercise between MODIS and Model AOD. Please clarify. Can the MODIS AOD be extracted at some of the stations and compared with Model data? It has also ben inferred in previous studies that MODIS data overpredicted AOD for regions predominant with dust (see Remer et al. 2005). Please take this into account while validation of the model. (Remer LA, Kaufman YJ, Tanré D, Matto S, Chu DA, Martins JV, Li RR, Ichoku C, Levy RC, Kleidman RG, Eck TF, Vermote E, Holben BN (2005) The MODIS aerosol algorithm, products, and validation. J Atmos Sci 62:947–973. https://doi.org/10.1175/JAS3385.1)

*As reviewer 1 has pointed out, we have limited aerosol observations for validation of this dust event. We have included the two MODIS aerosol retrievals during this event that had the best available domain coverage. In the discussion of the MODIS data we cited that the retrievals have an uncertainly of ~20% over land and 10-15% over water. We have added citation of Remer et al. (2005) and noted potential MODIS AOD overestimation in high dust loading environments.*

*While the MODIS AOD is useful for a qualitative comparison of the UAE and Saudi dust plumes, the data is quite patchy and covers only a portion of the domain. We have also noted that the modeled dust plumes in the RAMS simulations are slightly displaced compared to the corresponding high AOD plumes in the MODIS overpasses. These factors are prohibitive towards producing a meaningful quantitative comparison. However, visual qualitative comparisons reveal that the RAMS Ginoux and Walker simulations generate dust plumes in the region of the observed plumes. Further, the modeled plumes have AODs in the 1.5-2.5 range across the bulk of the plumes, which is very similar to the range of AOD seen in the MODIS data. While the MODIS data may have uncertainties up to 20% over land, the retrieved high AOD values are co-located with dense plumes seen in the visible imagery in Figure 4 and denote these plumes as being substantial dust events. As such, this event is worth examining in the model with respect to the potential variability in radiative effects due to different specification of dust erodible fraction.*

*In addition, we have interpolated the MODIS pixels to the location of the Mezaira AERONET site and added these point observations to the MODIS AOD figure. The interpolated MODIS AOD values from both overpasses are lower than the AERONET values, but are still indicative of a substantial dust event. As we note in the manuscript, the MODIS data is being interpolated to a point location in an area with a tight gradient in AOD and in the vicinity of missing pixels. As such, we suspect the interpolation tends to under-represent the high AOD at the indicated times compared to AERONET.*

2. A large underestimation is seen between model and AERONET AOD. What could be the reason for this? It will be nice if the authors could provide a quantitative validation, including bias and normalized mean error. How much is the uncertainty in AERONET AOD for regions predominant with dust? I suggest strengthening this Section by providing information from any available literature study as well. One of such studies, I recently found is by Kokkalis et al.,

(2018). Long-Term Ground-Based Measurements of Aerosol Optical Depth over Kuwait City. Remote Sensing, 10, 1807; DOI:10.3390/rs1011180710.

*The main point in providing the grid point comparisons of AOD is to generally demonstrate the presence of an intense dust plume in the area in both the observations and the model. As discussed in the paper, grid point comparisons, while potentially useful, can be deceptive when making comparisons in areas of tight gradients and areas where simulated features such as dust plumes are reasonably represented in the model but are slightly displaced compared to the observed location. Here, the underestimation in the model compared to AERONET is largely due to the fact that the model generates a dust plume over the UAE / Persian Gulf region that is slightly displaced to the east. Further, in the Walker simulation, there's a substantial gradient in dust AOD along the edges of the plume. As shown in the AERONET figure, a simulated in-plume grid point time series to the east of the Mezaira location does indeed reveal the passage of an intense dust plume. Such comparisons can be useful, but need to be cautiously interpreted. We have added some details from the Kokkalis et al. paper that help shed light on AODs that represent a mean background state for this region as well as dust storm AOD values.*

3. Also, why "Ginoux'" is larger than the "Walker" (refer to Figure 7c)? Please include some discussion on this.

*We note that figure 7c is a time series for a single grid point, so any spatial displacement between simulations can produce somewhat deceptive differences at single locations. The simulated UAE plume is displaced a bit to the east of the Mezaira location shown in the time series. As noted in the text, the Walker simulation lofts dust in more precise locations and then transports those with the wind. As such, the Walker dust plume is narrow and somewhat displaced from the Mezaira location. The Ginoux simulations have lower erodible fraction than the Walker dust locations, but the Ginoux sources cover a much larger area. As such, the Ginoux plume near Mezaira is more broadly dispersed but with a lower maximum AOD compared to the plume in the Walker simulation. We have added discussion of these differences and note that these differences need to be considered when interpreting time series of grid point comparisons.*

4. How much is the difference between the simulated dust concentration from NAAPS and that from RAMS and WRF? I suggest the authors discuss this as they provide NAAPS dust concentration.

*We agreed with reviewer 1 that inclusion of the NAAPS model snapshot does not offer much contribution to the paper since this is a comparison to an operational model and not real data. As such, we have removed the NAAPS figure panel and discussion from the paper.*

5. How much is the expected uncertainty in your model values for radiative impacts?

*There is not a general uncertainty that can be assigned to the radiation parameterization in the model. The RAMS radiation model physics predicts the radiative fluxes based on its radiative transfer equations that consider the presence of aerosols for this simulated event (Harrington 1997; Stokowski 2005).*

6. I suggest comparing the radiative implications, such as radiative cooling/heating during

daytime and nighttime with observational data.

*Observations of vertical profiles of radiative cooling/heating rates in and out of the dust plumes are not available. However, Stokowski (2005) demonstrated that RAMS is able to reasonably represent radiative heating associated with dust layers. Further, we have added discussion regarding the dust-induced changes in radiative fluxes and radiative heating/cooling and compared the RAMS simulated trends to those in Slingo et al. (2006) and Marsham et al. (2016). Both our results and those in the cited papers reveal a shortwave cooling trend at the surface due to dust as well as a counter balancing increase in radiative heating within the surface based dust layer. Both our results and the cited papers address the increase in radiative heating rates with respect to changes in the radiative flux divergence associated with attenuation of shortwave radiation by dense dust layers.*

7. Refer to Figure 10f: Is this for Total LW fluxes? Or for total radiative fluxes (SW+LW)? Please check.

*Figure 10 displays the mean profiles at night. As such, there are no shortwave contributions and only longwave fluxes are to be considered.*

---

## Author Response (AR2)

Co-Editor Decision: Publish subject to minor revisions (review by editor) (24 Jun 2019)
by Jui-Yuan Christine Chiu

**Reviewer comments below are in standard black font, while the author responses are in blue italic font for contrast.**

Comments to the Author:
Dear Stephen et al.,

Thanks very much for the revised manuscript and detailed responses. I am happy with your revisions, but do wish to see that the manuscript includes a bit more discussions on the implications of these model runs.

*In the specific responses below we note that we have included additional discussion in the conclusions section regarding the implications of our simulations, potential dust/radiation impacts on mesoscale phenomena, how these dust lofting simulations and analysis compare with other studies of dust radiative effects, and what we need in the future to better evaluate our models.*

Specifically, you summarize findings in the abstract, but I feel there is a lack of ending/conclusions. After all, what is the implication of these findings in terms of scientific understanding, model improvement, and observational needs for the future?

*We have added a paragraph in the conclusions section that addresses the need for more precise dust sources databases as well as the need for more in situ aerosol observations in these dust prone areas of the world in order to better evaluate predictions of dust lofting, transport, and radiative responses. We have also included additional summary statements to the abstract that mentions dust related aspects that are discussed in more detail in the body of the manuscript.*

The manuscript is partly motivated by the fact that few studies focus on the Arabian Peninsula, compared to the Sahara and East Asia (as mentioned in your Introduction). How are your findings different from what we knew about the other regions? Providing some thoughts/discussions (or compare/contrast) on it in one paragraph will greatly enhance the significance of the manuscript.

*We have added discussion in the conclusions section to provide an overarching comparison to previous dust lofting studies. We note in the conclusions that few previous studies have focused on the Arabian Peninsula due to few in situ dust and field study observations in the region. Further, the Saharan dust and East Asian dust studies have often been a focus due to their association with long range oceanic transport. We briefly summarize the differences in dust radiative effects on the surface between elevated dust plumes involved in long range transport and more locally based plumes that remain linked to the surface.*

Additionally, you may wish to reword a bit about Line 31, Page 3, since I don't think you have really addressed the impact on costal mesoscale features in this manuscript.

*We have removed the reference to coastal mesoscale features from this section on Page 3, and instead, have mentioned such features in the conclusions with regards to potential implications of dust radiative impacts and thermodynamic feedbacks.*

Finally, please state which collection of MODIS AOD was used in the study. If it is collection 6, Rob Levy's paper may be more appropriate than Remer et al. (2005).

*We used MODIS AOD collection 6.1. As such, we have removed the citation to Remer et al. (2005) and have added reference to collection 6.1 in the text. We already had the reference to Levy et al. (2013), and will thus, retain that reference.*

Please feel free to let me know if you have any question. I look forward to receiving your revisions soon.

Cheers
Christine

Non-public comments to the Author:
Please add space between numbers and units throughout the manuscript.

*We have added spaces between numbers and units.*

*We have also added the definition of "erodible fraction" to the abstract.*

[revised manuscript text omitted]